# Tuning self-renewal in the Arabidopsis stomatal lineage by hormone and nutrient regulation of asymmetric cell division

Yan Gong[1], Julien Alassimone[1†], Rachel Varnau[1], Nidhi Sharma[2], Lily S Cheung[3], Dominique C Bergmann[1,2]*

[1]Department of Biology, Stanford University, Stanford, United States; [2]Howard Hughes Medical Institute, Stanford University, Stanford, United States; [3]School of Chemical and Biomolecular Engineering, Georgia Institute of Technology, Atlanta, United States

**Abstract** Asymmetric and self-renewing divisions build and pattern tissues. In the Arabidopsis stomatal lineage, asymmetric cell divisions, guided by polarly localized cortical proteins, generate most cells on the leaf surface. Systemic and environmental signals modify tissue development, but the mechanisms by which plants incorporate such cues to regulate asymmetric divisions are elusive. In a screen for modulators of cell polarity, we identified *CONSTITUTIVE TRIPLE RESPONSE1*, a negative regulator of ethylene signaling. We subsequently revealed antagonistic impacts of ethylene and glucose signaling on the self-renewing capacity of stomatal lineage stem cells. Quantitative analysis of cell polarity and fate dynamics showed that developmental information may be encoded in both the spatial and temporal asymmetries of polarity proteins. These results provide a framework for a mechanistic understanding of how nutritional status and environmental factors tune stem-cell behavior in the stomatal lineage, ultimately enabling flexibility in leaf size and cell-type composition.

*For correspondence:
bergmann@stanford.edu

Present address: †
Plantpathology, Institute for Integrative Biology, ETH Zürich, Zürich, Switzerland

## Introduction

The cellular composition of a tissue defines its structure and functions. Tissue-embedded stem cells can control cell fate and distribution by modulating their division behaviors, often using symmetric cell divisions (SCDs) to renew stem-cell capacity and asymmetric cell divisions (ACDs) to diversify daughter cell fates (*De Smet and Beeckman, 2011*; *Losick et al., 2011*; *Morrison and Kimble, 2006*; *Motohashi and Asakura, 2014*). Tuning the relative proportion of ACDs to SCDs can modulate tissue size and organ composition in response to changes in the external and internal environment. For example, by shifting the ratio of ACDs to SCDs, intestinal stem cells in *Drosophila melanogaster* can resize the intestine in response to food availability (*O'Brien et al., 2011*), and in rat brains, stem cells are able to replenish differentiated neurons during stroke recovery (*Zhang et al., 2004*).

Cell polarity, the restricted localization of proteins, organelles, and activities to one region of the cell, is often linked to ACDs. Cell polarity can precede a division and dictate division orientation, thereby affecting daughter cell size and fate asymmetries, often through differential inheritance of specific materials (*Knoblich, 2001*; *Muroyama and Bergmann, 2019*). Although less studied, post-divisional polarity is also important, particularly in situations where cells undergo successive rounds of ACDs. Here, polarity must either be maintained or regenerated at each ACD. When the degree of polarity is not sufficient to ensure differential segregation of proteins to one daughter, it can trigger a developmental switch from ACDs to SCDs (and subsequent

differentiation), as was demonstrated for PAR proteins in *Caenorhabditis elegans* embryo development (*Hubatsch et al., 2019*).

The stomatal lineage in the epidermis of leaves is an excellent model to study how cell polarity and division behaviors interface with developmental and physiological flexibility. In Arabidopsis, the stomatal lineage produces two essential cell types, stomatal guard cells and pavement cells (*Figure 1A*). At any given time during development, stomatal lineages at different developmental stages can be found dispersed across the surface of a leaf. These lineages are initiated by ACDs that produce meristemoids and stomatal lineage ground cells (SLGCs). Successive ACDs in either meristemoids (amplifying divisions) or SLGCs (spacing divisions) are self-renewing. Terminal differentiation coincides with the SCD, and subsequent differentiation, of a guard mother cell (GMC) into guard cells. Altering the balance of differentiation and self-renewal (approximated by the SCD/ACD ratio) in the stomatal lineage changes the size, cellular composition, and pattern of the epidermis (*Bergmann and Sack, 2007*; *Vatén et al., 2018*). Because the epidermis largely determines leaf size (*Gonzalez et al., 2012*; *Vaseva et al., 2018*), SCD/ACD ratio also influences overall leaf properties. These stomatal lineage divisions are often downstream of systemic and environmental cues (*Engineer et al., 2014*; *Lau et al., 2018*; *Lee et al., 2017*; *Schroeder et al., 2001*). For example, a recent analysis of cytokinin hormone signaling showed that regulating the ability of SLGCs to undergo spacing ACDs contributes to developmental flexibility (*Vatén et al., 2018*).

Several proteins play crucial roles in determining cell fates and division behaviors in the stomatal lineage, including transcription factors (*Kanaoka et al., 2008*; *MacAlister et al., 2007*; *Ohashi-Ito and Bergmann, 2006*; *Pillitteri et al., 2007*), secreted peptide ligands, and cell surface receptors that mediate cell–cell communication (*Bergmann et al., 2004*; *Hara et al., 2007*; *Hunt and Gray, 2009*; *Nadeau and Sack, 2002*; *Qi et al., 2017*; *Shpak et al., 2005*). The transcription factor SPEECHLESS (SPCH) initiates ACDs, and its expression is maintained briefly in both daughter cells, then preferentially lost in the SLGC (*MacAlister et al., 2007*). Downstream targets of SPCH include 'polarity proteins': BREAKING OF ASYMMETRY IN THE STOMATAL LINEAGE (BASL), BREVIS RADIX-LIKE 2 (BRXL2), and POLAR LOCALIZATION DURING ASYMMETRIC DIVISION AND REDISTRIBUTION (POLAR) (*Dong et al., 2009*; *Lau et al., 2014*; *Pillitteri et al., 2011*; *Rowe et al., 2019*). These polarity proteins localize to cortical crescents and are required for ensuring the size and fate asymmetries of the ACD (*Dong et al., 2009*; *Houbaert et al., 2018*; *Pillitteri et al., 2011*; *Rowe et al., 2019*; *Zhang et al., 2016*; *Zhang et al., 2015*). Each of these polarity proteins can physically interact with signaling kinases and potentially act as scaffolds to ensure the kinases are active in the appropriate cell types and subcellular locations (*Houbaert et al., 2018*; *Marhava et al., 2018*; *Zhang et al., 2015*). The scaffolded kinases include MITOGEN ACTIVATED PROTEIN KINASES (MAPKs), Arabidopsis SHAGGY-LIKE kinases (ATSKs), and AGC kinases—all multifunctional kinases capable of mediating both developmental and environmental signals, thus potentially linking cell polarity to flexible and tunable development.

It is largely unknown how polarity proteins and their clients polarize during stomatal ACDs, and how polarity is linked to self-renewing capacity. From a genetic screen to identify regulators of cell polarity in the stomatal lineage, we found a mutation in *CONSTITUTIVE TRIPLE RESPONSE (CTR1)*, a core component of the ethylene signaling pathway, that resulted in overall depolarization of BRXL2. To understand the connection between CTR1 and BRXL2 polarity, and how the change of BRXL2 polarity affected leaf development, we created quantitative polarity analysis tools (*Gong et al., 2021*) and long-term tissue-wide lineage tracing methods. We discovered that ethylene and glucose signaling, respectively involved in environmental and nutritional pathways, antagonistically regulate the balance of asymmetric and symmetric cell divisions in the stomatal lineage. Additionally, we uncovered a new interaction between the two cells resulting from an ACD, where the temporal dynamics of BRXL2 polarity in an SLGC was linked to the self-renewing capacity of its sister meristemoid. Together, these results reveal previously underappreciated mechanisms that tune stem-cell behavior in the stomatal lineage.

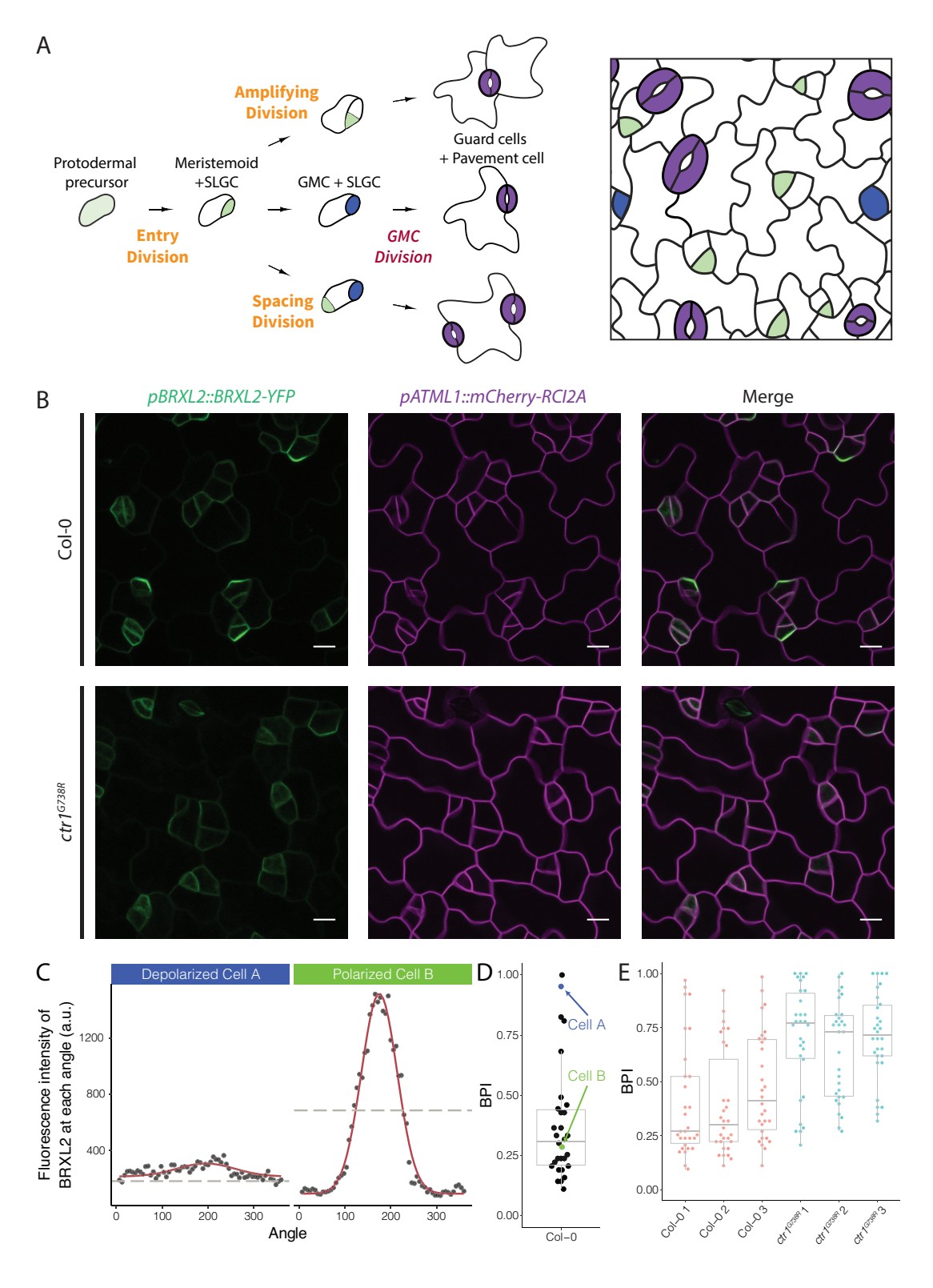

**Figure 1.** Quantitative analysis of BRXL2-YFP reporter during stomatal lineage divisions reveals reduction in polar localization in the loss-of-function mutant *ctr1*[G738R]. (**A**) Schematic diagram of stomatal lineage (left) and organization of leaf epidermis (right). From dispersed protodermal precursors, each asymmetric cell division (ACD) produces a small meristemoid (green) and a large stomatal lineage ground cell (SLGC, white). The meristemoid can self-renew by undergoing amplifying ACD(s) or differentiate into guard mother cells (GMCs, blue). Each GMC divides symmetrically to produce paired

*Figure 1 continued on next page*

*Figure 1 continued*

guard cells (purple). The SLGC can also undergo another ACD (spacing division) or differentiate into a pavement cell. Multiple stomatal lineages are initiated and undergo divisions and differentiation in a dispersed and asynchronized fashion. (B) BRXL2 localization epidermal in cells of 4 dpg Col-0 (top panels) and *ctr1^G738R* (bottom panels) cotyledons. *pBRXL2::BRXL2-YFP* (left), *pATML1::RCI2A-mCherry* (middle), and merged (right) are shown separately. (C) Output of POME measurement of depolarized (cell A, left) and polarized BRXL2 (cell B, right). Fluorescence intensity measurements of BRXL2 at each angle are plotted in black dots, and the nonlinear regression models per each cell are plotted in red. (D) POME quantification of BRXL2 polarity index (BPI) in Col-0 (n = 30 cells). Each point represents a BPI score calculated from the BRXL2 cortical localization pattern of one cell (details in Materials and methods and *Gong et al., 2021*). (E) Output of POME quantification of BRXL2 polarity in 4 dpg Col-0 and *ctr1^G738R* cotyledons (n = 30 cells/genotype, three replicates, statistical analysis reported in *Figure 1—figure supplement 3*). Scale bar in (B), 10 μm.

The online version of this article includes the following figure supplement(s) for figure 1:

**Figure supplement 1.** Molecular description of *ctr1^G738R* and other alleles, and whole-plant phenotypes resulting from *ctr1* mutants, artificial microRNA knockdown, and shoot epidermal-only expression of *CTR1*.

**Figure supplement 2.** Time-lapse imaging of BRXL2 dynamics during stomatal lineage divisions and additional quantification of BRXL2 polarity in Col-0 and *ctr1^G738R*.

**Figure supplement 3.** Display of full statistical tests among genotypes and replicates in *Figure 1E*.

# Results

## Loss of CTR1 reduces the fraction of stomatal lineage cells with polarized BRXL2-YFP

To identify regulators of cell polarity in Arabidopsis stomatal lineage, we performed a microscope-based screen for mutations that affected the subcellular localization of BRXL2. We screened ethylmethanesulfonate (EMS)-mutagenized seedlings expressing a native promoter-driven, yellow fluorescent protein (YFP) tagged, BRXL2 reporter (*pBRXL2::BRXL2-YFP*, *Figure 1B*, top panels). Among recovered mutants, we were particularly intrigued by a line where the BRXL2-YFP signal was mostly depolarized (*Figure 1B*, bottom panels). Through mapping and cloning by sequencing, we found this recessive mutant contained a G to A mutation in the coding region sequence of *CTR1*. CTR1 is a Raf-like kinase that couples with ethylene receptors, and its activity is required to inhibit the downstream ethylene signaling cascade (*Huang et al., 2003*). The mutation we found is predicted to cause a glycine to arginine substitution at position 738 in the kinase domain (*Figure 1—figure supplement 1A*), and thus, we will refer to this allele as *ctr1^G738R*. Like the previously reported amorphic allele *ctr1-1* (*Kieber et al., 1993*) and the hypomorphic allele *ctr1-btk* (*Ikeda et al., 2009*), *ctr1^G738R* shows a strong constitutive ethylene response phenotype at the seedling level (*Figure 1—figure supplement 1B*). The *ctr1^G738R* mutant also displayed a small leaf phenotype in older plants, similar to *ctr1-1* and stronger than *ctr1-btk*. This phenotype is consistent with leaf epidermal cells undergoing fewer stem-cell-like ACDs and an overall decreased number of cells in the leaf epidermis (*Figure 1—figure supplement 1C*).

To confirm that the disruption of *CTR1* caused the reduction in BRXL2 polarity, we introduced the *pBRXL2::BRXL2-YFP* reporter into the established *ctr1-btk* and *ctr1-1* mutants. We found that *ctr1-btk* and *ctr1-1* also caused different degrees of BRXL2 depolarization (*Figure 2A–D*). We also created a heteroallelic combination of *ctr1^G738R* with *ctr1-btk*. The F1 progeny of a cross between *ctr1^G738R pBRXL2::BRXL2-YFP* and *ctr1-btk* displayed an intermediate disruption in BRXL2 polarity, as well as an intermediate ethylene response phenotype (*Figure 2E*). These results confirm *CTR1* as the causal locus, but because *ctr1^G738R* (like *ctr1-1*) has severe developmental defects, we wanted to eliminate the possibility that the BRXL2 polarity disruption phenotype was a secondary consequence of broad and excessive ethylene signaling. We generated an artificial microRNA (amiRNA) knock-down line targeting *CTR1* under a stomatal lineage-specific promoter (*pTMM::amiRNA-CTR1*). *pTMM::amiRNA-CTR1* lines did not show a strong ethylene response at the seedling stage (*Figure 1—figure supplement 1D*), but did show reduced polarity of BRXL2 in the stomatal lineage (*Figure 2F*). Additionally, shoot epidermal-only expression of CTR1 (*pATML1::CTR1*) was sufficient to rescue the reduced BRXL2 polarity phenotype and the small leaf phenotype of *ctr1^G738R* (*Figure 1—figure supplement 1E–I*). These rescue results support a direct role of CTR1 in the epidermis

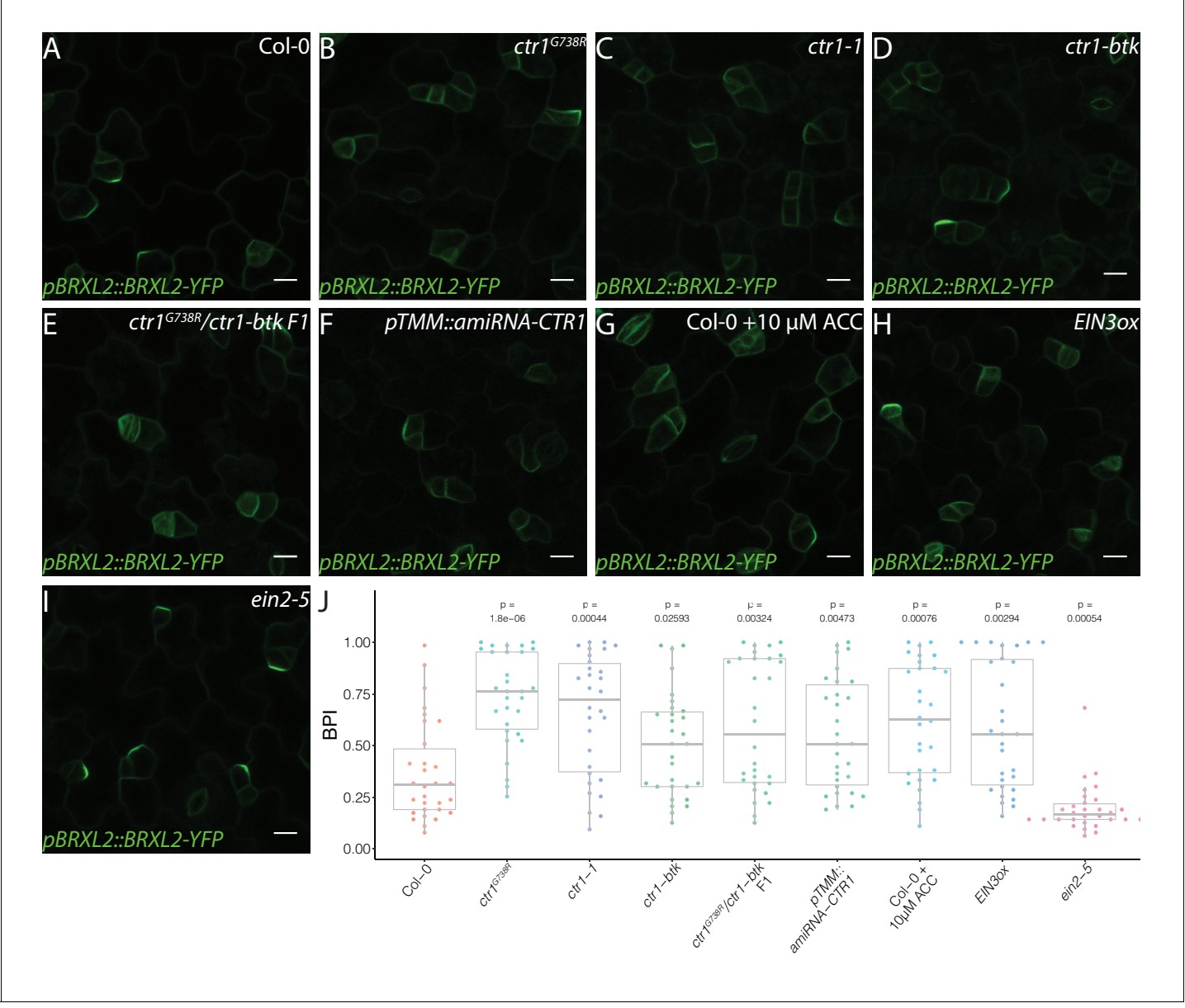

**Figure 2.** Active ethylene signaling decreases BRXL2 polarity. (**A–I**) BRXL2 localization pattern in 4 dpg cotyledons from different ethylene mutants or treatment (*pBRXL2::BRXL2-YFP* in green). (**A**) Col-0, (**B**) *ctr1^{G738R}*, (**C**) *ctr1-1*, (**D**) *ctr1-btk*, (**E**) *ctr1^{G738R}/ctr-btk* F1, (**F**) *pTMM::amiRNA-CTR1*, (**G**) 10 μM 1-aminocyclopropane-1-carboxylic acid-treated Col-0, (**H**) *EIN3ox*, and (**I**) *ein2-5*. (**J**) POME quantifications of BRXL2 polarity index from (**A–I**) conditions (n = 30 cells/genotype). All p-values are calculated by Mann–Whitney test. Scale bars in (**A–I**), 10 μm.
The online version of this article includes the following figure supplement(s) for figure 2:

**Figure supplement 1.** 1-Aminocyclopropane-1-carboxylic acid (ACC) does not affect BRXL2 polarity in ethylene-insensitive mutants.

and reinforce previous work showing that the epidermis drives organ growth in leaves (*Marcotrigiano, 2010*; *Vaseva et al., 2018*).

# Time-lapse imaging combined with quantitative polarity analysis reveals nuances of stomatal lineage cell polarity

Among *ctr1* mutant lines, BRXL2-YFP was consistently depolarized in most, but not all, cells. This suggested that *CTR1* is not absolutely essential for BRXL2 polarity establishment and emphasized that to accurately interpret the role of *CTR1* we needed a full picture of BRXL2 polarity dynamics throughout the stomatal lineage. We therefore monitored the dynamics of BRXL2-YFP during development of the entire cotyledon over 2 days in 40 min intervals. Consistent with previous reports (*Rowe et al., 2019*), a polar crescent of BRXL2 is visible before asymmetric division (*Figure 1—figure supplement 2A*, top-left panels; *Video 1*), and after division, this crescent is inherited by the large daughter cell (SLGC). Time lapse, however, revealed two additional features of BRXL2 behavior: first, BRXL2 persists in the SLGC for more than 8 hr after division (*Figure 1—figure supplement 2A*, top-left panels); and second, BRXL2 is still expressed in symmetrically dividing GMCs, but is depolarized in these cells (*Figure 1—figure supplement 2A*, top-right panels). These results suggest a correlation between the degree of BRXL2 polarity and cell identity. Thus, we applied our recently developed polarity measurement tool (POME, *Gong et al., 2021*) to quantify the distribution of BRXL2-YFP signal at the periphery of each cell. With POME, the pixel intensity of the BRXL2 reporter and an evenly distributed PM reporter are captured along the entire cell circumference, and their relative distributions were used to compute a 'polarity index' (*Figure 1C*; *Gong et al., 2021*). The BRXL2 polarity index (BPI) is defined as the fraction of measurements above the half maximum; it represents the fraction of the plasma membrane occupied by BRXL2 and ranges between 0 and 1. A BPI close to 0 represents a cell with highly polarized BRXL2, while a BPI of 1 represents a cell with completely depolarized BRXL2. As illustrated for Col-0 in *Figure 1D*, it is also possible to capture the distribution of BPI measurements in the population of BRXL2-expressing cells. Because of the tight correlation we found between cell identity and BPI, this 'snapshot' population measure can also be used to estimate the ratio of SCDs to ACDs occurring in the leaf epidermis.

We quantified the BPI of Col-0 and *ctr1* cotyledons at 4 days post germination (dpg). As expected, *ctr1^{G738R}* cotyledons showed a higher average BPI than Col-0 (*Figure 1E*), with fewer cells displaying low BPIs (highly polarized BRXL2). When we specifically analyzed the low BPI cells, we found no significant difference in crescent size or peak height between Col-0 and *ctr1^{G738R}* (*Figure 1—figure supplement 2B–D*). Through additional time-lapse imaging, we demonstrated that, like ACDs in Col-0, stomatal lineage ACDs in *ctr1^{G738R}* cotyledons were always preceded by the

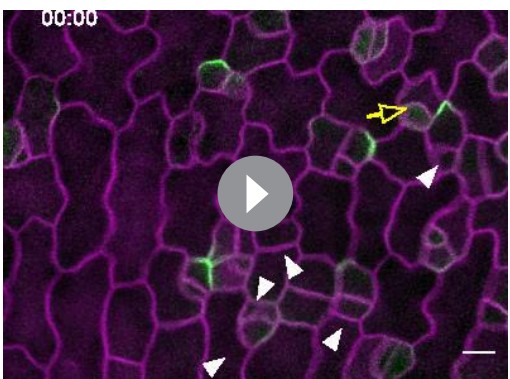

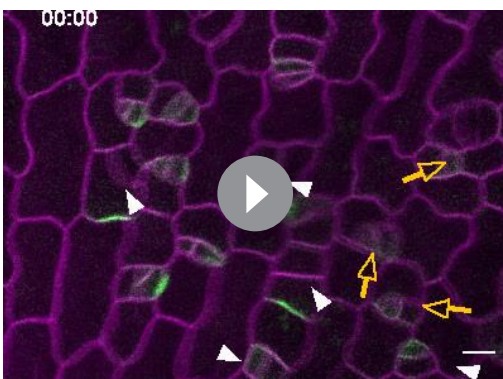

**Video 1.** BRXL2 dynamics in Col-0. Examples of BRXL2 dynamics (green, *pBRXL2::BRXL2-YFP*) during stomatal lineage asymmetric cell divisions (labeled by white arrowheads) and symmetric cell divisions (labeled by yellow arrows) on the abaxial surface of a 3 dpg wild-type Col-0 cotyledon. Cell membrane is labeled by *pATML1::RCI2A-mCherry* in magenta. White numbers in frames indicate hours:minutes relative to first frame. Scale bar, 10 μm.

https://elifesciences.org/articles/63335#video1

**Video 2.** BRXL2 dynamics in *ctr1^{G738R}*. Examples of BRXL2 dynamics (green, *pBRXL2::BRXL2-YFP*) during stomatal lineage asymmetric cell divisions (labeled by white arrowheads) and symmetric cell divisions (labeled by yellow arrows) on the abaxial surface of a 3 dpg *ctr1^{G738R}* cotyledon. Cell membrane is labeled by *pATML1::RCI2A-mCherry* in magenta. White numbers in frames indicate hours:minutes relative to first frame. Scale bar, 10 μm.

https://elifesciences.org/articles/63335#video2

appearance of polarized BRXL2 in the precursor cell (*Figure 1—figure supplement 2A*, bottom-left panels; *Video 2*).

These results raised a conundrum: if cells in *ctr1*$^{G738R}$ can polarize BRXL2, then how was *ctr1*$^{G738R}$ identified and mapped based on a disrupted BRXL2 polarity phenotype? Two potential explanations emerged, each reflecting the dynamic nature of polarity. First, because BRXL2 is polarized in cells undergoing ACDs and depolarized in cells undergoing SCDs, altering the ratios of these division types could decrease the proportion of cells with polarized BRXL2. This change might be detected as a population-level BRXL2 polarity decrease during the screen and suggests that the role of *CTR1* is primarily to maintain stem-cell capacity. Second, we had also observed that BRXL2-polarized crescents did not persist as long after ACDs in *ctr1*$^{G738R}$ compared to Col-0 (Figure 6A), and this reduction of BRXL2 persistence would also result in the appearance of relatively fewer polarized cells when observing BRXL2 polarity at a single timepoint. This result suggests several possible models of CTR1 action in regulating BRXL2 polarity and ACD, but also raises questions about post-divisional functions of polarity factors and how polarity is maintained.

## Disruption of *CTR1* results in diminished stem-cell capacity

To dissect the relationship between BRXL2 behavior with stem-cell capacity and with stomatal cell fate determination, we calculated the stomatal index (SI, ratio of stomata to all epidermal cells) of fully developed cotyledons of Col-0 and *ctr1* mutants at 14 dpg. Compared to Col-0, the SIs of different *ctr1* mutants were significantly elevated (*Figure 3A, B, E*, *Figure 3—figure supplement 1A–C*). Because of the flexible trajectory of the stomatal lineage (*Figure 1A*), increased SI has several possible origins: (1) an increase in cells entering the stomatal lineage (entry division), (2) an increase in secondary entry via SLGC spacing divisions, or (3) a decrease in meristemoid self-renewal by ACDs (amplifying divisions). To evaluate the contributions of these possibilities, we developed a whole-leaf-based lineage tracing method. We tracked the developmental progression of all epidermal cells in Col-0 and *ctr1*$^{G738R}$ cotyledons within a 48 hr time window (3–5 dpg) and captured the developmental progression of more than 500 pairs of meristemoids and SLGCs per genotype (*Figure 3F*). Strikingly, the percentage of amplifying ACDs from *ctr1*$^{G738R}$ cotyledons was significantly reduced, while the percentage of spacing ACDs was similar to Col-0 (*Figure 3G*). When individual meristemoids were tracked, we found many underwent fewer rounds of amplifying ACDs during 48 hr in *ctr1*$^{G738R}$ compared to Col-0 (*Figure 3H*).

Fewer amplifying ACDs should result in fewer epidermal cells; this was confirmed by comparing total cell numbers of whole Col-0 and *ctr1*$^{G738R}$ cotyledons at 3, 4, and 5 dpg. At 3 dpg, *ctr1*$^{G738R}$ and Col-0 cotyledons had similar numbers of cells, suggesting a similar level of stomatal entry divisions. However, by 5 dpg, Col-0 had accumulated about 30% more cells than *ctr1*$^{G738R}$ (*Figure 3I*). Together, these results suggest that meristemoids prematurely exit stem-cell divisions in *ctr1*$^{G738R}$ plants, thereby elevating the SCD/ACD ratio and ultimately generating fewer epidermal cells, a higher SI, and smaller leaves.

## Ethylene signaling regulates polarity protein complex and stomatal lineage development

*CTR1* is best known as a negative regulator of ethylene signaling, but it also is enmeshed in crosstalk with other signaling pathways. Ethylene's ability to affect stomatal lineage development was previously noted (*Serna and Fenoll, 1996*), but the details of the regulation were not clear. To test whether ethylene, in general, affects stomatal lineage ACD/SCD decisions, we treated *pBRXL2::BRXL2-YFP*-expressing seedlings with the ethylene precursor 1-aminocyclopropane-1-carboxylic acid (ACC). A 10 µM ACC treatment significantly increased average BPI at 4 dpg and SI at 14 dpg relative to mock-treated controls (*Figure 2G, J*, *Figure 3C, E*). Upon ethylene reception, one of the core elements of the ethylene pathway, ETHYLENE INSENSITIVE 2 (EIN2), is cleaved, translocates to the nucleus, and activates the transcription factors EIN3 and its homolog ETHYLENE INSENSITIVE LIKE 1 (EIL1) to mediate ethylene signaling (*Alonso et al., 1999*; *An et al., 2010*; *Chang et al., 2013*; *Chao et al., 1997*; *Qiao et al., 2012*; *Wen et al., 2012*). CTR1 acts as a negative regulator by phosphorylating EIN2 and preventing its cleavage (*Qiao et al., 2012*; *Wen et al., 2012*). In plants overexpressing EIN3 (*EIN3ox*) (*Chao et al., 1997*), the average BPI and SI were higher than the average BPI and SI in Col-0 (*Figure 2H, J*, *Figure 3E*, *Figure 3—figure supplement 1D*). Conversely, a

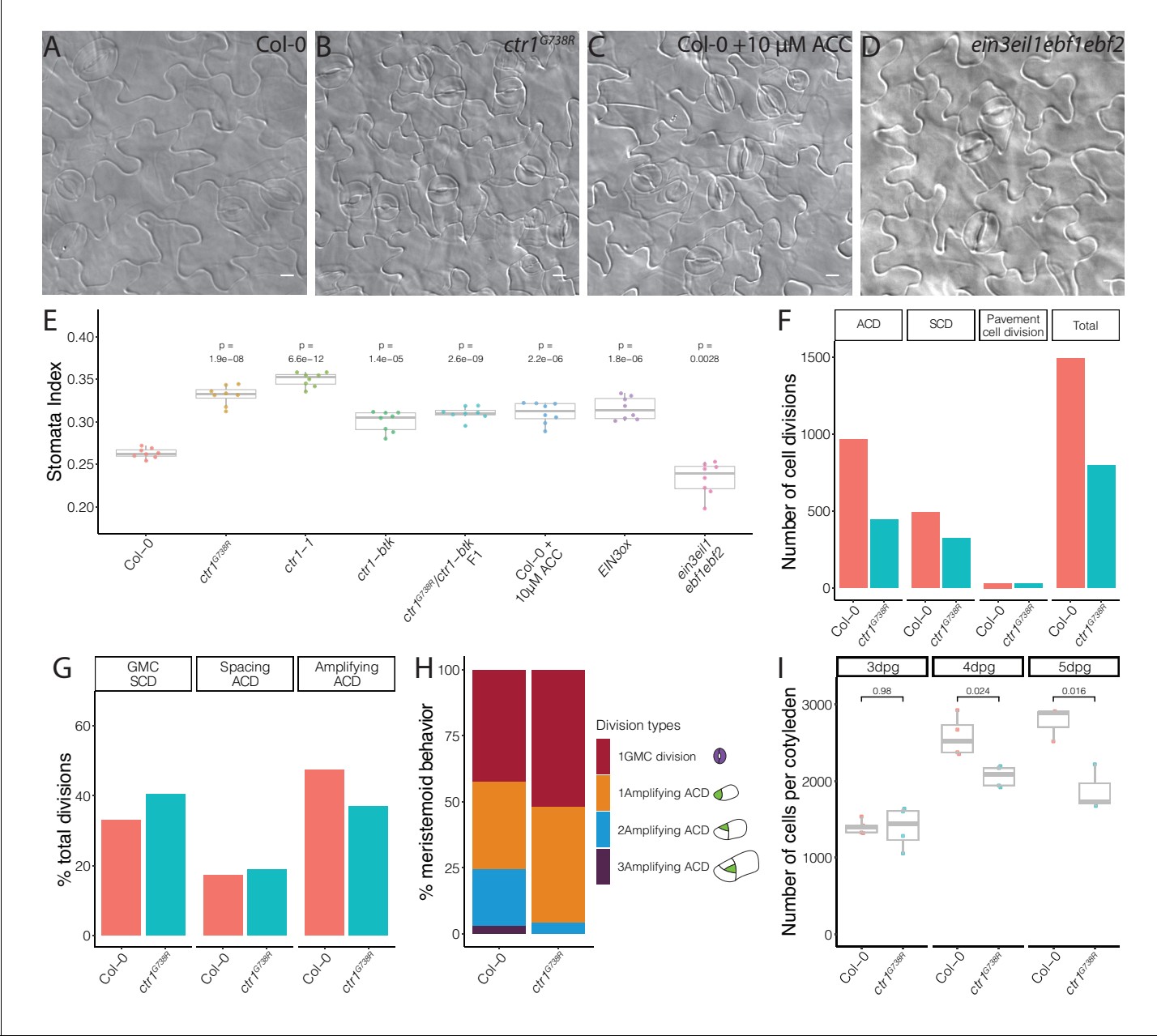

**Figure 3.** Ethylene signaling modulates the symmetric cell division/asymmetric cell division balance during leaf epidermal development. (A–D) Differential interference contrast (DIC) images of (A) Col-0, (B) ctr1^G738R, (C) 10 µM 1-aminocyclopropane-1-carboxylic acid (ACC)-treated Col-0, and (D) ein3eil1ebf1ebf2 14 dpg cotyledons grown on ½ Murashige and Skoog media without sugar. (E) Quantification of the stomatal index of Col-0, ACC-treated Col-0, and selected ethylene signaling mutants (n = 8/genotype). (F–H) Results of tracing epidermal cell lineages in Col-0 and ctr1^G738R cotyledons. (F) Total numbers of different division types tracked. (G) Percentage of each division type among total divisions. (H) Quantification of number of divisions the tracked meristemoids undergo, accompanied by cartoon representation (n > 500 cells/genotype). (I) Total abaxial epidermal cell number in Col-0 and ctr1^G738R cotyledons from 3 to 5 dpg (n = 3–5 cotyledons/genotype/day). All p-values are calculated by Student's t-test due to the small sample sizes. Scale bars in (A–D), 10 µm.

The online version of this article includes the following figure supplement(s) for figure 3:

**Figure supplement 1.** DIC images of cotyledon epidermis of ethylene signaling mutants.

slight decrease of BPI and SI was observed in the ethylene-insensitive mutants *ein2-5* (*Alonso et al., 1999*) and the quadruple mutant *ein3eil1ebf1ebf2*, where EBF (EIN3-binding F-BOX) proteins were eliminated to approximate complete lack of ethylene response (*An et al., 2010*, *Figure 2I, J*,

*Figure 3D, E*). To test whether CTR1 acts primarily through the EIN2/EIN3 core pathway to regulate stomatal stem-cell divisions, we treated *ein2-5* single mutants and *ein3eil1ebf1ebf2* quadruple mutants with 10 µM ACC for 4 days. In these ethylene-insensitive backgrounds, ACC treatment did not affect BRXL2 polarity (*Figure 2—figure supplement 1*), suggesting that CTR1 acts via the canonical ethylene signaling pathway to modulate meristemoid division behaviors, regulate stomatal density, and limit leaf growth.

These genetic and pharmacological perturbations of ethylene signaling indicate that ethylene signaling disrupts BRXL2 polarity primarily through shifting cell identity from meristemoids (which polarize BRXL2) to GMCs (which do not). These results demonstrate how a systemic signal alters tissue-level development and suggest that BPI may be an easily scorable proxy for stem-cell potential, a phenotype that was tedious and labor-intensive to measure previously (*Vatén et al., 2018*).

## Glucose signaling antagonizes ethylene signaling and enhances amplifying divisions

For BPI to be generally useful as an estimate of stomatal stem-cell potential, we needed to identify situations where stem-cell potential was enhanced to provide a counterpoint to the repression we observed in *ctr1*. Previous studies found that ethylene and sugar signaling can act in opposition, and so we hypothesized that if sugar signaling also acted in the stomatal lineage, altered availability or perception of sugar would be reflected in the BPI (*Haydon et al., 2017*; *Karve et al., 2012*; *Yanagisawa et al., 2003*; *Zhou et al., 1998*). Indeed, addition of sucrose to the media resulted in a dose-dependent decrease in average BPI in both *ctr1^{G738R}* and Col-0 cotyledons (*Figure 4A–D, K*, *Figure 4—figure supplement 1*).

The effect of sucrose on BPI could be due to its role as a nutrient or a signal. In Arabidopsis, sucrose is the major transport form of sugar and the major energy source for sink tissues where it is further broken down into glucose and fructose to provide energy for cell metabolism, growth, and divisions. Glucose also acts as a signaling molecule during plant development and responds to environmental cues via a hexokinase (HXK)-mediated pathway (*Eveland and Jackson, 2012*). We therefore tested the ability of metabolizable, non-metabolizable, signaling-active, and signaling-inactive forms of sugars to change BPI in Col-0 and *ctr1^{G738R}*. Addition of glucose, but not fructose, decreased average BPI (*Figure 4E–H, L*). The glucose signaling-inactive analog, 3-*O*-methyl-D-glucose (3-OMG) (*Cortès et al., 2003*), failed to alter BPI (*Figure 4I, J, L*), suggesting that it is glucose signaling, rather than cellular energy status, that affects stomatal lineage progression. To test whether these treatments specifically targeted BRXL2 or whether BRXL2-YFP reports a more general effect on cell polarity, we tested the effects of glucose and ethylene on BASL polarity using a *pBASL::YFP-BASL* reporter line (*Figure 4—figure supplement 2A*; *Rowe et al., 2019*). Like BRXL2 polarity, BASL polarity was significantly enhanced with 2% glucose treatment and reduced with 10 µM ACC treatment (*Figure 4—figure supplement 2B–E*). These results suggest that glucose and ethylene impact cell polarity in the stomatal lineage more generally.

We then tested whether the stomatal lineage response to ethylene was mediated by the polarity proteins themselves. In stomatal development, *BRXL2* is genetically redundant with other *BRX* family members, and the BRX family appears to physically interact with and be mutually dependent on BASL (*Rowe et al., 2019*). Therefore, to most cleanly assay hormone response in the absence of polarity protein activity, we treated the *BASL* null mutant (*basl-2*) with ACC. 10 µM ACC can still increase the stomatal index in *basl-2*, although the degree of increase is less than that in Col-0 (*Figure 4—figure supplement 2F*). These results are consistent with polarity proteins playing a minor role in directly mediating ACC response.

To confirm our model that a decrease in BPI upon glucose treatment reflected an increase in stem-cell potential, we tracked division types of all epidermal cells in time courses. Consistent with its ability to decrease average BPI in treated plants, 2% glucose treatment promoted amplifying ACDs in both Col-0 and *ctr1^{G738R}* (*Figure 5A*). Additionally, the SI of *ctr1^{G738R}* grown in the presence of 2% glucose was shifted back to the wild-type level (*Figure 5B, Figure 5—figure supplement 1*), consistent with glucose-antagonizing ethylene signaling and boosting the ability of meristemoids to undergo amplifying ACDs.

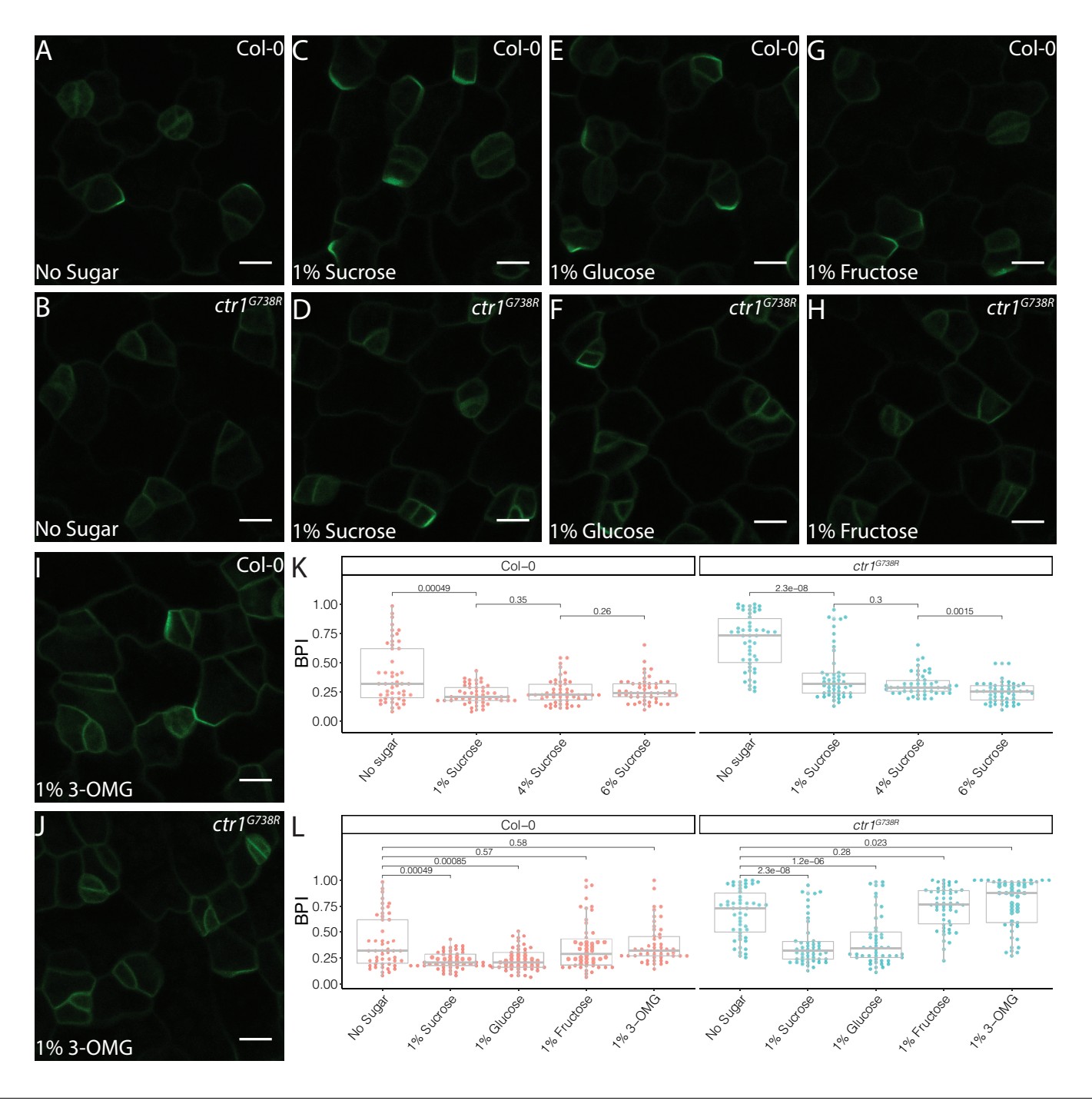

**Figure 4.** Sucrose and glucose signaling increase BRXL2 polarity. (A–J) BRXL2 localization pattern in 4 dpg cotyledons grown on ½ Murashige and Skoog (MS) plates with various sugars (*pBRXL2::BRXL2-YFP* in green). (A, C, E, G, and I) Col-0 and (B, D, F, H, and J) *ctr1^G738R*. (A, B) No sugar, (C, D) 1% sucrose, (E, F) 1% fructose, and (I, J) 1% 3-*O*-methyl-D-glucose. (K) BRXL2 polarity index (BPI) quantification of Col-0 and *ctr1^G738R* grown on ½ MS plates with different sucrose concentrations (n = 50 cells/genotype). (L) BPI quantification in 4 dpg Col-0 and *ctr1^G738R* seedlings growing on ½ MS plates with various sugars (n = 50 cells/genotype). The same BPI measurements of Col-0 and *ctr1^G738R* from no sugar and 1% sucrose treatment are included in (K) and (L) for easier visual comparison. All p-values are calculated by Mann–Whitney test. Scale bars in (A–J), 10 μm.

The online version of this article includes the following figure supplement(s) for figure 4:

**Figure supplement 1.** BRXL2 localization of Col-0 and *ctr1^G738R* grown on high levels of sucrose.

**Figure supplement 2.** BASL polarity is altered by glucose and ethylene signaling.

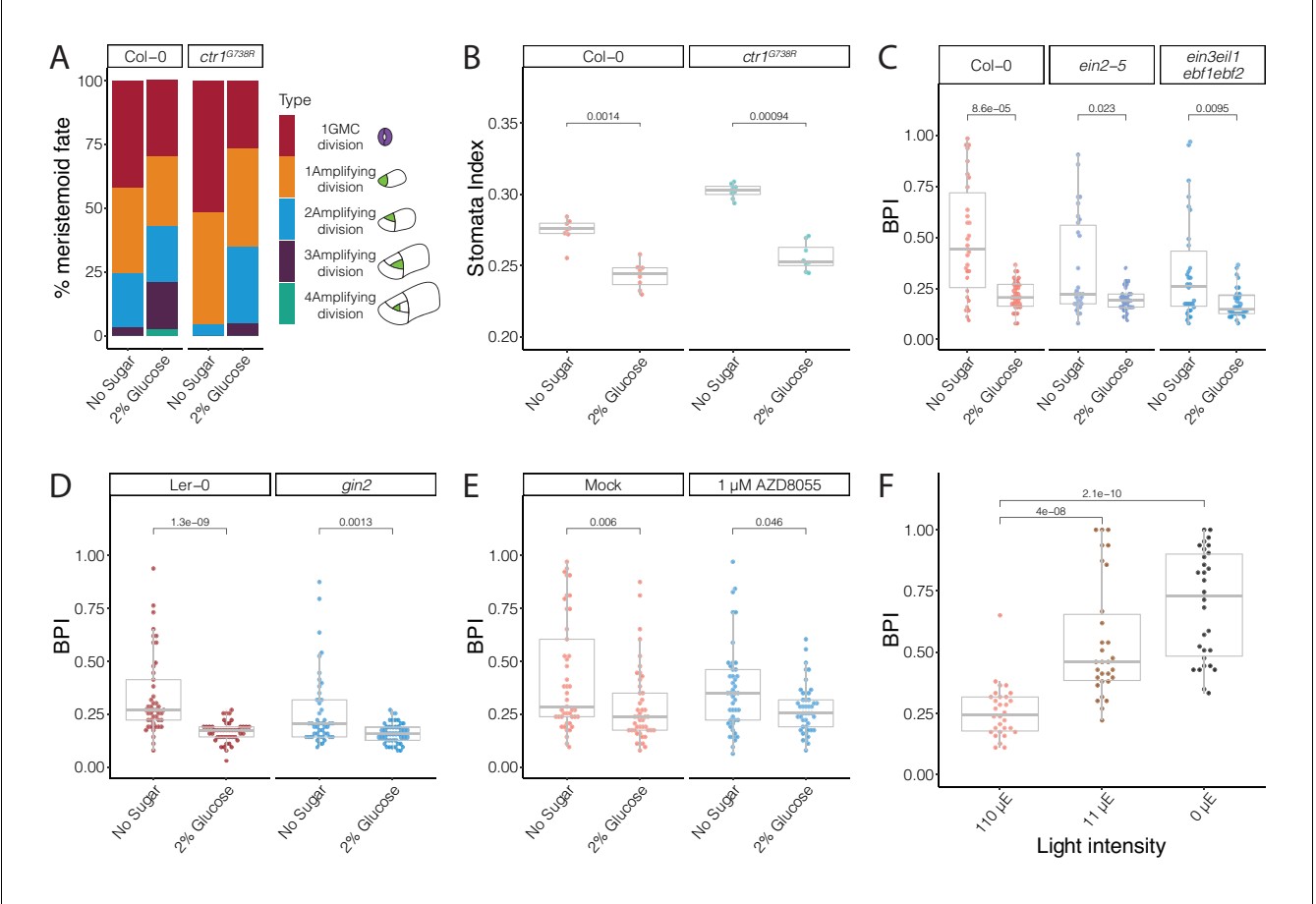

**Figure 5.** Glucose promotes amplifying asymmetric cell divisions independent of ethylene signaling and HXK1- or TOR-mediated glucose signaling pathways. (**A**) Cartoon of types of divisions meristemoids undergo, and fraction of each type in Col-0, *ctr1$^{G738R}$*, Col-0 with 2% glucose treatment, and *ctr1$^{G738R}$* with 2% glucose treatment (n > 500 cells/condition). Data for the no sugar condition of Col-0 and *ctr1$^{G738R}$* are also reported in *Figure 3H*. (**B**) Stomatal index of 14 dpg Col-0 and *ctr1$^{G738R}$* growing on ½ Murashige and Skoog (MS) plates or 2% glucose ½ MS plates (n = 8/condition). (**C**) BRXL2 polarity index (BPI) quantification of 4.5 dpg abaxial cotyledons of Col-0, *ein2-5,* and *ein3eil1ebf1ebf2* grown on ½ MS plates or 2% glucose ½ MS plates (n = 30 cells/genotype). (**D**) BPI quantification of 4 dpg abaxial cotyledons of Ler-0 and *gin2* grown on ½ MS plates or 2% glucose ½ MS plates (n = 45 cells/genotype). (**E**) BPI quantification in 4 dpg Col-0 treated with TOR inhibitor AZD8055 and/or 2% glucose (n = 30 cells/genotype). (**F**) BPI quantification of true leaves in 9 dpg Col-0 seedlings 110 μE normal light condition for 7 days and then transferred to different low light intensity conditions for 48 hr (n = 30 cells/genotype). All p-values are calculated by Mann–Whitney test.

The online version of this article includes the following figure supplement(s) for figure 5:

**Figure supplement 1.** DIC images of cotyledons from seedlings grown on ½ Murashige and Skoog (MS) media with or without 2% glucose.

**Figure supplement 2.** BRXL2 localization pattern in Col-0 and ethylene-insensitive mutants under different light and sugar treatment regimes.

**Figure supplement 3.** Glucose control of BRXL2 polarity is independent of HXK1 signaling.

**Figure supplement 4.** Glucose control of BRXL2 polarity is independent of TOR signaling.

**Figure supplement 5.** Glucose increases BRXL2 polarity in Col-0 and *ein3eil1ebf1ebf2* true leaves grown under low-light condition.

## Glucose control of stomatal differentiation is independent of ethylene signaling, HXK1 signaling, or TOR signaling

The relationship between glucose and ethylene signaling has been explored in detail in other contexts, leading to the model that active HKX-1-mediated glucose signaling promotes EIN3 degradation and reduces ethylene signaling activity (*Yanagisawa et al., 2003*). To test if glucose regulates stomatal divisions through EIN3 inhibition, we compared BPIs in the ethylene-insensitive mutants *ein2-5* and *ein3eil1ebf1ebf2* grown with and without glucose in the media. Both mutants had significantly reduced BPIs with 2% glucose treatment at 4.5 dpg (*Figure 5C*, *Figure 5—figure supplement*

*2A–F*), suggesting that the influence of glucose in stomatal divisions is independent of the core components of ethylene signaling.

To test another mechanism by which glucose signaling is connected to stomatal lineage development, we quantified the BPI in previously established HXK1 loss-of-function mutants, *hxk1-3* (*Huang et al., 2015*) and *gin2* (*Moore et al., 2003*), in response to glucose treatment. When treated with 2% glucose, both mutants exhibited a decrease in average BPI similar to their corresponding wild-type controls, Col-0 and Ler-0 (*Figure 5D, Figure 5—figure supplement 3A–J*), suggesting that glucose's regulation in stomatal development is not mediated by HXK1. Another glucose signaling pathway, the target of rapamycin (TOR) pathway, has been suggested to act downstream of sugar signaling as a major controller of plant growth-related processes, including meristem proliferation, leaf initiation, and cotyledon growth (*Li et al., 2017*; *Rexin et al., 2015*; *Xiong et al., 2013*). We treated *pBRXL2::BRXL2-YFP*-expressing seedlings grown with and without 2% glucose with 1 μM AZD-8055, an ATP-competitive inhibitor of TOR (*Montané and Menand, 2013*), for 2 days. Despite the presence of AZD-8055, glucose still decreased the average BPI (*Figure 5E, Figure 5—figure supplement 4*). Therefore, despite clear evidence that glucose signaling can influence stomatal lineage behaviors, we have been unable to link this regulation to known HXK1 or TOR-mediated sugar signaling pathways.

## Stomatal BPI can respond to physiological depletion of sugars

By experimentally adding sugars, we could modulate BPI and stomatal divisions. The critical question then becomes whether this is biologically relevant—do Arabidopsis leaf epidermal cells sense endogenous levels of glucose or sucrose and adjust the stomatal lineage to create leaves of appropriate cellular composition? To test this, we adapted an approach used in *Moraes et al., 2019* and reduced light intensity to limit the photosynthetic rate in seedlings. Because sugar is the primary product of photosynthesis, this experimental procedure serves to exhaust endogenous sugars in leaves. We transferred 7 dpg Col-0 seedings from our regular high-light intensity (110 μE) growth condition to low-light conditions (11 or 0 μE). After 48 hr, seedlings grown in low-light conditions showed increased BPIs in their leaves (*Figure 5F, Figure 5—figure supplement 2G–I*), indicating fewer ACDs took place. We could reverse the effect of low (11 μE) light on BPI in the true leaves of Col-0 by addition of 2% glucose, and this response also occurred in *ein3eil1ebf1ebf2* plants (*Figure 5—figure supplement 5*). Together, these results are consistent with sugar in the leaf providing feedback to coordinate epidermal development with photosynthesis.

## Post-division BRXL2 crescent is associated with meristemoid fate determinacy

In previous sections, we showed that *CTR1*, ethylene, and sugar signaling regulate the balance between stomatal lineage cells undergoing proliferative ACDs and differentiating SCDs, where the connection to BRXL2 polarity is largely indirect. However, in our examination of *ctr1*$^{G738R}$, we also noticed that polarized BRXL2 was in fewer SLGCs than it was in Col-0 and suspected that BRXL2 was less persistent in these cells. We monitored persistence of the polarized BRXL2 crescent directly through time-lapse imaging. The BRXL2 crescent was significantly less persistent in *ctr1*$^{G738R}$ than in Col-0 (*Figure 6A, B*, *Figure 6—figure supplement 1A*), and glucose increased persistence of BRXL2 crescent in both *ctr1*$^{G738R}$ and Col-0 seedlings (*Figure 6C*). In light of the observation that meristemoids in *ctr1*$^{G738R}$ underwent fewer amplifying ACDs (*Figure 3H*) and glucose promoted amplifying ACDs in both Col-0 and *ctr1*$^{G738R}$ (*Figure 5A*), a positive correlation emerges between the persistence of the post-division BRXL2 polarity complex in the SLGC and the self-renewal capacity of the meristemoid (the SLGC sister derived from the previous ACD).

Two plausible explanations for a non-cell-autonomous effect of BRXL2 persistence on its sister meristemoid's behavior are (1) that these cells communicate or (2) that by monitoring BRXL2 we witnessed a differentiation event already specified in their mother cell; for example, a change in SPCH activity that distinguishes cells with higher self-renewal potential and cells that will undergo a final ACD.

SPCH initiates all the ACDs in the stomata lineage (*MacAlister et al., 2007*) and directly binds the promoters of *BRXL2* and *BASL* in ChIP-Seq studies (*Lau et al., 2014*). BRXL2 crescent persistence, and the change of that persistence in response to ethylene and glucose treatment, therefore,

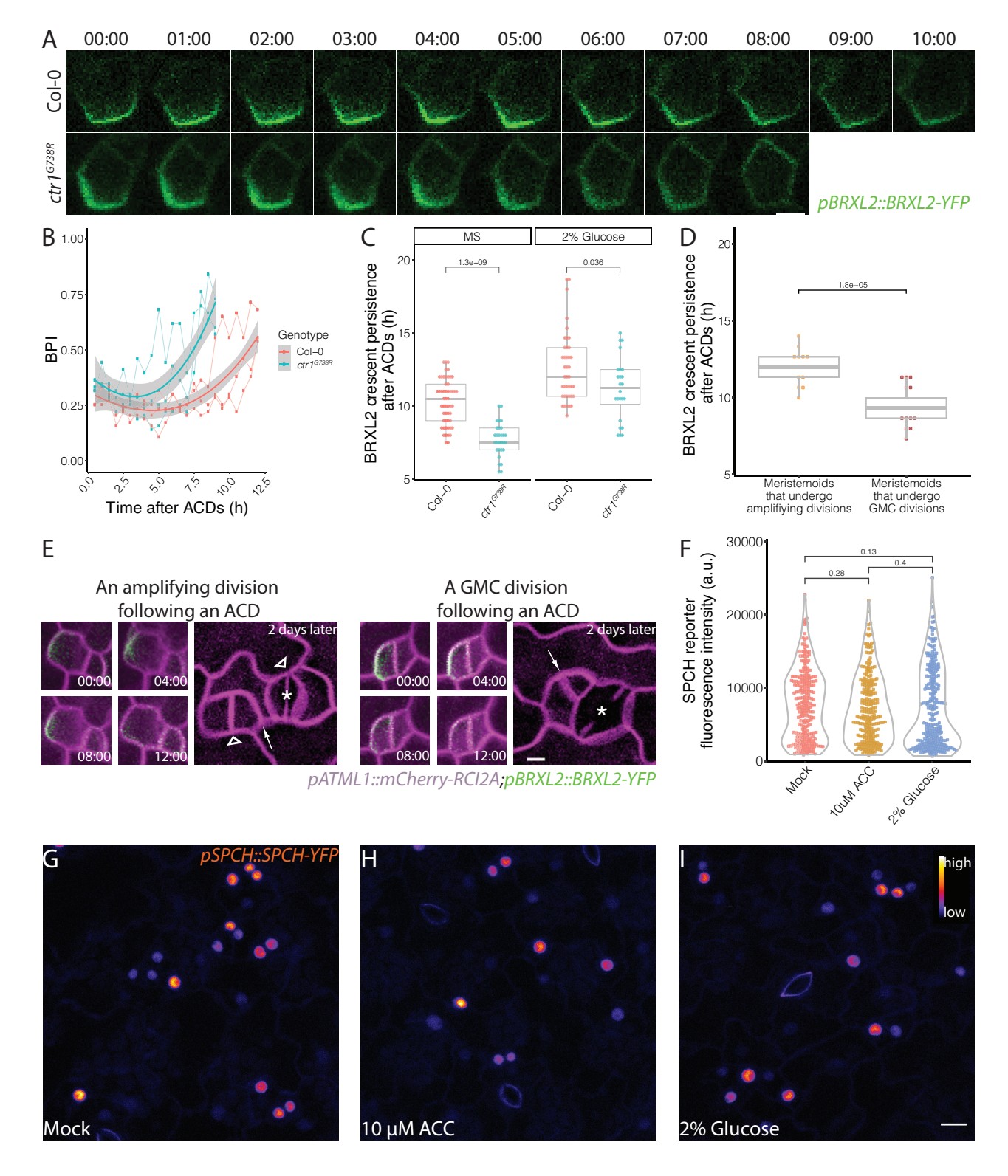

**Figure 6.** Post-division persistence of BRXL2 polarity is associated with meristemoid fate determinacy. (**A**) Time-lapse images of post-asymmetric cell division (ACD) BRXL2 dynamics from stomatal lineage ground cells (SLGCs) in Col-0 and *ctr1*[G738R] cotyledons (3 dpg). 00:00 (hours:minutes) marks cell plate formation. (**B**) Quantification of BRXL2 polarity dynamics after ACDs in Col-0 and *ctr1*[G738R]. Individual measurement per each cell (n = 3 cells per genotype) shown in thin lines and the respective trend per each genotype with 0.95 confidence interval is indicated as the thick line with gray band. (**C**)
*Figure 6 continued on next page*

*Figure 6 continued*

Persistence of BRXL2 post-ACD in SLGCs from 3 dpg Col-0 and *ctr1*$^{G738R}$ grown in ½ Murashige and Skoog (MS) media or ½ MS media with 2% glucose. (D) Relationship between persistence of BRXL2 in SLGCs and division behavior of their meristemoid sisters. (E) Examples of division behaviors quantified in (D). Time-lapse analysis of BRXL2 polarity in 3 dpg Col-0 cotyledons followed by lineage tracing. BRXL2 was imaged every 40 min for 16 hr, then plants returned to ½ MS plate for 48 hr, then reimaged to capture divisions and fate of the BRXL2-expressing cells. Different division types are marked with asterisks (GMC division), triangles (amplifying division), and arrows (spacing divisions). (F–I) Evidence that ethylene and glucose signaling does not affect SPCH level in individual stomatal lineage cells. (F) Quantification of gSPCH reporter fluorescence intensity at 4 dpg in (G) mock, (H) 10 µM 1-aminocyclopropane-1-carboxylic acid, and (I) 2% glucose-treated Col-0 cotyledons (n = 3 cotyledons/treatment; n > 120 cells/treatment). Lookup table Fire is used to false color gSPCH reporter intensity (color key in figure). All p-values are calculated by Mann–Whitney test. Scale bars in (A, E), 5 µm; (I), 10 µm.

The online version of this article includes the following figure supplement(s) for figure 6:

**Figure supplement 1.** Additional characterization of BRXL2 and SPCH dynamics during asymmetric cell divisions (ACDs).

might be directed by quantitative changes in SPCH expression in the cells undergoing ACD. We quantified SPCH protein levels in individual cells from 10 µM ACC, 2% glucose, and mock-treated Col-0 cotyledons by measuring the fluorescence intensity of a functional genomic SPCH reporter (*pSPCH::gSPCH-YFP* in *spch3*) (*Lopez-Anido et al., 2020*). We found no significant change in gSPCH-YFP levels upon treatment with either ACC or glucose (*Figure 6F–I*). We also examined gSPCH reporter dynamics during ACDs in these different treatment conditions and found no obvious changes in gSPCH reporter peak intensity, onset of expression, or post-divisional persistence (*Figure 6—figure supplement 1C, D*).

If BRXL2 crescent persistence in an SLGC is not coupled to the upstream (SPCH) transcriptional response, then this persistence is unlikely to be reflecting a decision made pre-ACD in the mother cell. To correlate BRXL2 crescent persistence post-ACD with the subsequent divisions and fates of the daughters, we performed detailed time-lapse imaging (16 hr, 40 min intervals) followed by a 48 hr time course in Col-0. We found that the post-ACD BRXL2 crescent persistence in SLGCs was predictive of sister meristemoid behavior. BRXL2 crescents persisted, on average, 2 hr longer when meristemoids underwent amplifying ACDs than when meristemoids divided symmetrically (*Figure 6D, E*). Interestingly, BRXL2 was not predictive of the behavior of the cell in which it resides. There was no significant difference between persistence in SLGCs that underwent spacing ACDs and those that did not (*Figure 6—figure supplement 1B*). Together, these results suggest that there must be communication between the sister cells resulting from an ACD. Hypotheses about the nature of this communication are presented in *Figure 7* and will be discussed below.

## Discussion

Plants respond to environmental stimuli by modifying their development. As the major organs of photosynthesis, leaves must regulate their size, position, and gas-exchange capacity to adapt and compete. Using genetics, live-cell imaging, lineage tracing, and quantitative image analysis of the simple, yet flexible, Arabidopsis stomatal lineage, we have been able to link ethylene and sugar signaling to the self-renewing capacity of epidermal stem cells. The immediate readout of ethylene and glucose antagonism is a shift in population of cells expressing polarized vs. depolarized BRXL2, and the ultimate readout is a change in the size and the cell-type composition of the leaf (*Figure 7A*). We also uncovered a surprising correlation between the ACD potential of meristemoids and the persistence of polarized BRXL2 in SLGCs (*Figure 7B*), suggesting active communication and coordination between these sister cells.

Finding *CTR1* alleles in screens for ACD regulators was not expected, and we considered whether there was a true role for ethylene signaling or whether we were seeing the effect of altering *CTR1* on other pathways. Genetic and pharmacological experiments (*Figures 2* and *3*), however, confirmed the participation of multiple ethylene signaling components in regulating stomatal lineage ACDs. Ethylene also inhibits stem-cell divisions in the shoot apical meristem of Arabidopsis (*Hamant et al., 2002*), revealing a common theme to regulation of stem-cell behaviors in the aerial

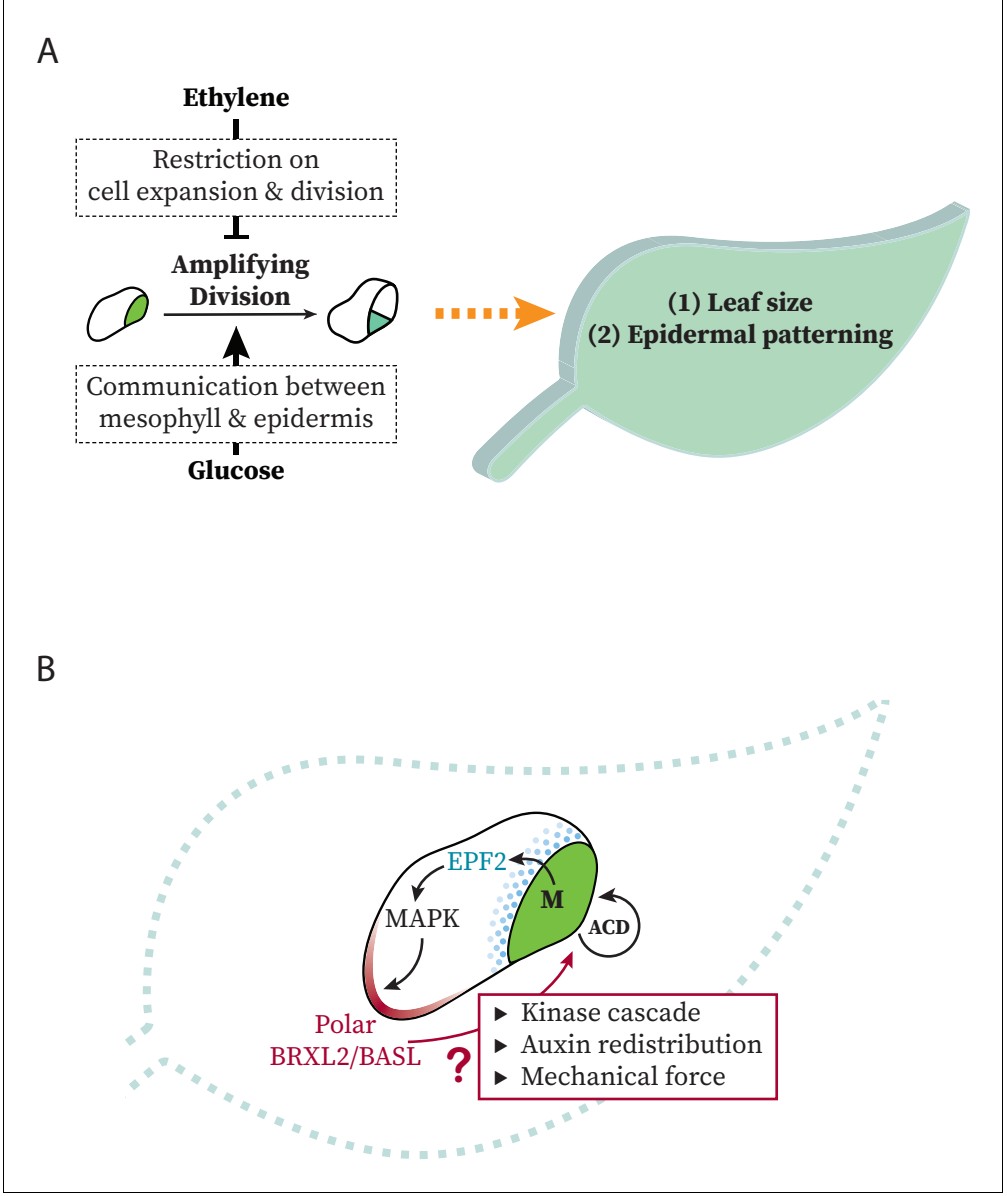

**Figure 7.** Models at organ and cell scales for connection between systemic signaling, cell polarity, stomatal stem-cell potential, and leaf growth. (**A**) Schematic representation of the regulation of ethylene and glucose on amplifying divisions in meristemoids. Potential regulatory mechanisms used by these signals are illustrated in dashed box. (**B**) Schematic representation of modes of communication possible between stomatal lineage ground cells and meristemoids during polarity and cell division control.

tissues. Ethylene is considered an 'aging' and a 'stress' hormone (*Iqbal et al., 2017*; *Schaller, 2012*) and regulates many different aspects of plant development, often through its cross-talk with auxin (*Muday et al., 2012*; *Strader et al., 2010*; *Vaseva et al., 2018*; *Wen et al., 2012*). In both root and leaf epidermal cells, for example, ethylene promotes local auxin biosynthesis. Elevated auxin levels then inhibit cell expansion and regulate plant growth (*Vaseva et al., 2018*). Elevated auxin signaling can also induce expression of ethylene biosynthesis genes (*Abel et al., 1995*; *Tsuchisaka and Theologis, 2004*), creating a feedback loop between these two hormones. In our case, ethylene might regulate stomatal lineage cell divisions through its feedback with auxin. Auxin has also been shown

to influence cell division and differentiation in the stomatal lineage, but there are some conflicting results, with *Le et al., 2014* suggesting that auxin promotes meristemoid ACDs and *Balcerowicz et al., 2014* concluding that auxin inhibits ACDs. The interplay of auxin and ethylene and the influence of auxin on all types of stomatal lineage divisions and fate transitions are beyond the scope of this paper, but will be an exciting future direction and may be answered more definitively by new tools like engineered TIR1 (*Uchida et al., 2018*) expressed in specific stomatal lineage cell types.

We did pursue the cross-talk between ethylene and sugar signaling, and found that higher levels of each resulted in opposite effects on BRXL2 persistence and stomatal lineage ACDs. Introduction of 2% glucose to growth media increased the stem-cell-like ACDs, even in ethylene-insensitive mutants, indicating that the epidermis is likely perceiving these signals independently. An effect of sugars on stomatal production was recently reported (*Han et al., 2020*). Their experimental conditions differ substantially from ours, however, making direct comparisons of results impossible, but both studies concur that sugars promote stomatal development. What information is sugar providing? Glucose and sucrose are the major products of photosynthesis produced in mesophyll cells, and recent work demonstrates multiple modes of communication between mesophyll and epidermis to coordinate these tissues for optimal growth and gas exchange (*Baillie and Fleming, 2020*; *Dow et al., 2017*; *Sugano et al., 2010*). It is attractive to consider mesophyll-derived sugar signaling as a way to promote growth and stomatal production, and future experiments could be designed to trace the source of sugar perceived by the stomatal lineage.

Perhaps our most interesting and surprising result was that temporal variations in BRXL2 persistence correlated with ACD potential. In particular, although BRXL2 is expressed in SLGCs, it was the behavior of the sister meristemoid that was affected. This made us consider what properties of the meristemoid are most important for that cell's behaviors (*Figure 7B*). Previous work considered the expression of the transcription factor SPCH as the key to modulating stomatal ACDs (*Lau et al., 2018*; *Simmons et al., 2019*; *Vatén et al., 2018*; *Zhang et al., 2016*). However, while SPCH is necessary for divisions, neither glucose nor ethylene signaling had a dose-dependent effect on SPCH levels. On the other hand, meristemoids vary in cell size, and ethylene and sugar have opposite effects on cell expansion. Thus, one possibility, parallel to the situation in the *C. elegans* germline, is that smaller meristemoids undergo fewer rounds of ACDs before committing to terminal differentiation. This hypothesis could also explain the difference in ethylene's effect on ACD potential in meristemoids and SLGCs. SLGCs are larger than meristemoids, thus ethylene-mediated repression of cell expansion will cause meristemoids to fall below the critical size threshold more often than SLGCs.

By what mechanisms could SLGCs influence their sister meristemoid, and how might persistent BRXL2 polarity drive this regulation? Signals from the meristemoid to the SLGC rely on the mobile peptide EPIDERMAL PATTERNING FACTOR2 (EPF2). Perception of EPF2 leads to higher MAPK activity in SLGCs (*Lee et al., 2015*). Elevated MAPK activity not only inhibits divisions (*Bergmann et al., 2004*; *Lampard et al., 2009*) but also enhances BASL polar localization (*Zhang et al., 2015*). Thus, BRXL2 polarity persistence in the SLGC could be a readout of the ACD potential of the sibling meristemoid. In this scenario, EPF2 secreted by the self-renewing meristemoid is perceived by the neighboring SLGC, stimulating the MAPK signaling pathway in that SLGC. Elevated MAPK activity then promotes nuclear export and cortical enrichment of BASL (*Zhang et al., 2015*), which in turn sustains BASL and BRXL2 polarity. When the meristemoid becomes a GMC, EPF2 is no longer produced and MAPK signaling is not activated in the neighboring SLGC. BASL is retained in the nucleus, and, without its partner, BRXL2 becomes depolarized.

We propose, however, that longer persistence of the polarity domain in the SLGC can also act as the source of a signal. This signal may be either chemical or mechanical (or both). For example, BASL and BRXL2 polar crescents precede local cell outgrowth (*Bringmann and Bergmann, 2017*; *Mansfield et al., 2018*) and differential expansion in the SLGCs could also serve as a division-promoting signal to the meristemoid. Such a mechanism has been shown to maintain tissue integrity among mechanically coupled cells (*Hamant and Haswell, 2017*). Polarity proteins BASL, BRX, and POLAR scaffold MAPKs, PAX, and BIN2 kinases, respectively (*Houbaert et al., 2018*; *Marhava et al., 2018*; *Zhang et al., 2015*). In many situations, including these three scaffold/kinase examples, scaffolding increases signaling output and creates extensive positive feedback on polarity

(*Houbaert et al., 2018*; *Marhava et al., 2018*; *Zhang et al., 2015*). Additionally, higher signal accumulation in auxin (MAPK, PAX) or brassinosteroid (BIN2) pathways can also lead to the production of mobile signals, and such signals have been connected to coordination of growth and stomatal lineage progression (*Houbaert et al., 2018*; *Kim et al., 2012*; *Le et al., 2014*). Distinguishing among these mechanisms will require new tools to specifically alter polarity crescent persistence in an otherwise unperturbed background, but advances in inducible degradation systems (*Faden et al., 2016*; *Sallee et al., 2018*) may enable these experiments in the future.

# Materials and methods

## Key resources table

| Reagent type (species) or resource | Designation | Source or reference | Identifiers | Additional information |
|---|---|---|---|---|
| Gene (*Arabidopsis thaliana*) | CTR1 | TAIR (https://www.arabidopsis.org/) | AT5G03730 | Receptor-coupled kinase involved in ethylene signaling |
| Gene (*Arabidopsis thaliana*) | BRXL2 | TAIR (https://www.arabidopsis.org/) | AT3G14000 | Polarity protein in the stomatal lineage |
| Gene (*Arabidopsis thaliana*) | BASL | TAIR (https://www.arabidopsis.org/) | AT5G60880 | Polarity protein in the stomatal lineage |
| Gene (*Arabidopsis thaliana*) | ATML1 | TAIR (https://www.arabidopsis.org/) | AT4G21750 | Homeobox transcription factor |
| Gene (*Arabidopsis thaliana*) | EIN2 | TAIR (https://www.arabidopsis.org/) | AT5G03280 | Component of ethylene signaling |
| Gene (*Arabidopsis thaliana*) | EIN3 | TAIR (https://www.arabidopsis.org/) | AT3G20770 | Transcriptional factor in ethylene signaling |
| Gene (*Arabidopsis thaliana*) | EIL1 | TAIR (https://www.arabidopsis.org/) | AT2G27050 | Transcriptional factor in ethylene signaling |
| Gene (*Arabidopsis thaliana*) | EBF1 | TAIR (https://www.arabidopsis.org/) | AT2G25490 | F-box protein involved in ethylene signaling |
| Gene (*Arabidopsis thaliana*) | EBF2 | TAIR (https://www.arabidopsis.org/) | AT5G25350 | F-box protein involved in ethylene signaling |
| Gene (*Arabidopsis thaliana*) | HXK1 | TAIR (https://www.arabidopsis.org/) | AT4G29130 | Hexokinase in the glucose signaling network |
| Gene (*Arabidopsis thaliana*) | SPCH | TAIR (https://www.arabidopsis.org/) | AT5G53210 | bHLH transcription factor involved in stomatal development |
| Gene (*Arabidopsis thaliana*) | TMM | TAIR (https://www.arabidopsis.org/) | AT1G80080 | LRR receptor-like protein involved in stomatal development |
| Strain, strain background (*Arabidopsis thaliana*) | Col-0 | ABRC | CS28166 | Wild-type Arabidopsis ecotype used in this study |
| Strain, strain background (*Arabidopsis thaliana*) | Ler-0 | ABRC | CS20 | Wild-type Arabidopsis ecotype used in this study |
| Strain, strain background (*Agrobacterium tumefaciens*) | GV3101 | Other | | Electrocompetent *A. tumefaciens* |
| Genetic reagent (*Arabidopsis thaliana*) | ctr1-1 | *Kieber et al., 1993*; DOI:10.1016/0092-8674(93)90119-b | | |
| Genetic reagent (*Arabidopsis thaliana*) | ctr1-btk | *Ikeda et al., 2009*; DOI:10.3389/fpls.2017.00475 | | |
| Genetic reagent (*Arabidopsis thaliana*) | ein3eil1ebf1ebf2 | *Sugano et al., 2010*; DOI: 10.1105/tpc.110.076588 | | |

*Continued on next page*

*Continued*

| Reagent type (species) or resource | Designation | Source or reference | Identifiers | Additional information |
|---|---|---|---|---|
| Genetic reagent (*Arabidopsis thaliana*) | *ein2-5* | *Alonso et al., 1999*; DOI:10.1126/science.284.5423.2148 | | |
| Genetic reagent (*Arabidopsis thaliana*) | *gin2* | *Moore et al., 2003*; DOI:10.1126/science.1080585 | | |
| Genetic reagent (*Arabidopsis thaliana*) | *hxk1-3* | *Huang et al., 2015*; DOI:10.3389/fpls.2015.00851 | | |
| Genetic reagent (*Arabidopsis thaliana*) | *EIN3ox* | *Chao et al., 1997*; DOI:10.1016/s0092-8674(00)80300–1 | | |
| Genetic reagent (*Arabidopsis thaliana*) | *ctr1$^{G738R}$* | This paper | | A *ctr1* (in Col-0, *Arabidopsis thaliana*) mutant harboring a point mutation; first introduced in *Figure 1* and fully described in *Figure 1—figure supplement 1*; request to DCB Laboratory, Stanford, USA |
| Genetic reagent (*Arabidopsis thaliana*) | *pTMM::amiRNA-CTR1* | This paper | | Transgenic line expressing an artificial microRNA (amiRNA) knockdown construct targeting *CTR1* under the stomatal lineage-specific promoter *TMM*; first introduced in *Figure 2*; request to DCB Laboratory, Stanford, USA |
| Genetic reagent (*Arabidopsis thaliana*) | *pATML1::CTR1; ctr1$^{G738R}$* | This paper | | Transgenic line expressing *CTR1* under the epidermal-specific promoter *ATLML1* in the *ctr1$^{G738R}$* mutant background; first introduced in *Figure 1—figure supplement 1*; request to DCB Laboratory, Stanford, USA |
| Genetic reagent (*Arabidopsis thaliana*) | *pBRXL2::BRXL2-YFP; pATML1::mCherry-RCI2A* | *Rowe et al., 2019*; DOI: 10.1101/614636 | | |
| Genetic reagent (*Arabidopsis thaliana*) | *pBASL::YFP-BASL; pATML1::mCherry-RCI2A* | *Rowe et al., 2019*; DOI: 10.1101/614636 | | |
| Genetic reagent (*Arabidopsis thaliana*) | *pSPCH::gSPCH-YFP; spch3* | *Lopez-Anido et al., 2020*; DOI: 10.1101/2020.09.08.288498 | | |
| Chemical compound, drug | Propidium iodide | Thermo Fisher | Thermo Fisher: P3566 | Dye for staining intercellular space in Arabidopsis |
| Chemical compound, drug | AZD-8055 | Fisher Scientific | Fisher Scientific: 50-101-5840 | ATP-competitive inhibitor of TOR |
| Chemical compound, drug | 1-Aminocyclopropane-1-carboxylic acid | Sigma | Sigma: A3903-100MG | Ethylene precursor |
| Chemical compound, drug | Glucose | Sigma | Sigma: G7021-1KG | |
| Chemical compound, drug | 3-*O*-Methyl-D-glucopyranose | Sigma | Sigma: M4849-10G | Non-metabolizable sugar |
| Chemical compound, drug | Sucrose | Sigma | Sigma: S3929-1KG | |

*Continued on next page*

*Continued*

| Reagent type (species) or resource | Designation | Source or reference | Identifiers | Additional information |
|---|---|---|---|---|
| Chemical compound, drug | Fructose | Sigma | Sigma: 1040071000 | |
| Commercial assay or kit | RNeasy plant mini kit | QIAGEN | QIAGEN: 74104 | |
| Commercial assay or kit | iScript cDNA synthesis kit | Bio-Rad | Bio-Rad: 170-8891 | |
| Commercial assay or kit | Ssoadvanced Universal SYBR Green Supermix | Bio-Rad | Bio-Rad: 172-5274 | |
| Software, algorithm | Leica Application Suite X | Leica | | Version: 3.5.2.18963 |
| Software, algorithm | FIJI | *Schindelin et al., 2012*; DOI:10.1038/nmeth.2019 | | Version: 2.0.0-rc-69/1.52 p |
| Software, algorithm | POME | *Gong et al., 2021*; DOI:10.1111/nph.17165 | | Version:1.0.0 |
| Software, algorithm | R | https://www.R-project.org/ | | Version: 4.0.1 |
| Software, algorithm | TrackMate | *Tinevez et al., 2017*; DOI: 10.1016/j.ymeth.2016.09.016 | | Version: 6.0.1 |
| Software, algorithm | RStudio | https://rstudio.com | | Version: 1.3.959 |

## Plant material and growth conditions

All Arabidopsis lines used in this study are in Col-0 background except for *gin-2* (Ler-0), and sources of previously reported mutants and transgenic lines are listed in the Key Resources Table. Newly generated lines include *pTMM::CTR1amiRNA* and *pATML1::CTR1; ctr1^{G738R}*.

All Arabidopsis seeds were surface-sterilized by bleach or 75% ethanol and stratified for 2 days. After stratification, seedlings were vertically grown on ½ Murashige and Skoog (MS) media with 1% agar for 3–14 days under long-day condition (16 hr light/8 hr dark at 22°C) and moderate-intensity full-spectrum light (110 µE) unless noted otherwise.

## Identification and map-based cloning of CTR1

Col-0 seeds homozygous for the reporter *pBRXL2::BRXL-YFP* were mutagenized with ethylmethane-sulfonate (EMS). Seedlings from M2 families were screened individually at 3–5 dpg for loss of polar YFP localization on a Leica SP5 confocal microscope. Candidate mutants were then backcrossed, and F2 progeny with and without the mutant phenotype (>50 plants of each type) were pooled, sequenced, and analyzed as in *Wachsman et al., 2017* with the exception that identification of potentially causal single-nucleotide polymorphisms was done by sequence comparisons among the many mutants rather than relative to a reference line.

## Vector construction and plant transformation

To generate *pTMM::amiRNA-CTR1*, an artificial microRNA sequence targeting *CTR1* was designed with the Web MicroRNA Designer platform (http://wmd3.weigelworld.org) (*Schwab et al., 2006*), engineered with the pRS300 plasmid with the *TMM* promoter, and cloned into the binary vector R4pGWB401 (*Nakagawa et al., 2008*). *pATML1::CTR1* was generated by cloning CTR1 coding sequence from cDNA into pENTR and combining with ML1 promoter sequences in binary vector R4pGWB401. Primers used to generate these two constructs are listed in the primers section below. Transgenic plants were then generated by Agrobacterium-mediated transformation (*Clough and Bent, 1998*), and transgenic seedlings were selected on ½ MS plates with 50 µM kanamycin.

## Drug treatments

To assay the influence of different types and concentrations of sugar on BPI or SI, filter-sterilized 40% sucrose, 20% glucose, 20% fructose, or 20% 3-OMG water solutions were prepared as stock solutions and added to the sterilized ½ MS media with 1% agar to make ½ MS sugar treatment plates. Similarly, ACC was dissolved in water and filter-sterilized to create the 100 mM ACC stock

solution. This ACC stock solution was then added to the sterilized ½ MS media with 1% agar to make ½ MS ACC treatment plates. Surface-sterilized and -stratified Arabidopsis seeds were then plated on these plates and vertically grown for 3–14 days under long-day condition for the respective experiments. For AZD-8055 experiments, AZD-8055 was dissolved in DMSO to create 1 mM stock solution. AZD-8055 stock solution or DMSO was then added to the sterilized ½ MS media with 1% agar to make ½ MS AZD-8055 or mock treatment plates. Surface-sterilized and -stratified Arabidopsis seeds were then plated on regular ½ MS plates and vertically grown for 2 days prior to being transferred to AZD-8055 or mock treatment plates and vertically grown two more days before image acquisition. For time-lapse experiments, ½ MS solution was supplemented with 2% glucose prior to loading seedling into the chamber, and all flow-through solution contained the same glucose concentration.

## Microscopy and image acquisition

All fluorescence imaging experiments were performed on a Leica SP5 or Leica SP8 confocal microscope with HyD detectors using 40× NA1.1 water objective with image size 1024*1024 and digital zoom from 1× to 2×.

For time-lapse experiments, 3 dpg seedlings were mounted in a custom imaging chamber filled with ½ MS solution (*Davies and Bergmann, 2014*). Laser settings for each reporter, except the membrane marker (*pATML1::mCherry-RCI2A*), were adjusted to avoid over-saturation. For the time-lapse experiments reported in this study, there was a 30–45 min interval between each image stack capture. For the reoccurring time-lapse experiments, seedlings were imaged in the time-lapse chamber for 16 hr with the protocol stated above. After imaging, seedlings were removed from the imaging chamber and returned to MS-agar plates (with appropriate supplements) for 8 hr under standard light and temperatures. The same epidermal surface from the same plant was reimaged using the same time-lapse protocol each successive day from 3 to 5 dpg. The three sets of time-lapse images were combined together, with the time between recordings noted, to create a time-lapse covering about 64 hr of development.

For time-course (lineage tracing) experiments, where images of the same whole epidermis were acquired every 24 hr from 3 to 5 dpg, each seedling was carefully mounted on a slide with vacuum grease outlining the border of the cover slide. In this setting, vacuum grease provides mechanical support to avoid crushing the seedlings. After each image acquisition, seedlings were carefully unmounted from the slide and returned to the ½ MS plate and normal growth condition until the next image acquisition. For the analysis of these time-course (lineage tracing) experiments, refer to the 'Lineage tracing analysis' section.

All raw fluorescence image Z-stacks were projected with Sum Slices in FIJI unless noted otherwise. For all time-lapse images, drift was corrected using the Correct 3D Drift plugin (*Parslow et al., 2014*) prior to any further analysis.

For SI counting, seedlings were collected at 14 dpg. Samples were cleared with 7:1 solution (7:1 ethanol:acetic acid), treated with 1 N potassium hydroxide solution, rinsed in water, and then mounted in Hoyer's solution. Individual leaves were then imaged with a Leica DM6B microscope with 20× NA0.7 air objective in differential contrast interference mode.

## Image quantification

For POME measurements of BRXL2 polarity in different mutants and treatments, florescence images of BRXL2 and membrane marker or staining in these conditions were acquired with 40× water objective and 2× digital zoom. Three to five individual images were captured from the same region of cotyledon from different seedlings. For each individual image, relative brightness of BRXL2 reporter was used to select the 10 brightest cells, and POME was used to measure BRXL2 distribution along the cell membrane in each of the selected cells. This selection was made as a way to capture cells that had recently divided either symmetrically or asymmetrically and was the fairest comparison of BRXL2 polarity across different mutants and treatments. With POME, the cortex of each cell was divided into 63 portions, and the fluorescence intensity of BRXL2 reporter in each portion was measured and reported. Details of BRXL2 measurement with POME are available in *Gong et al., 2021*. BPI is defined as the fraction of measurements above the half maximum. To calculate the BPI for each cell, the maximum BRXL2 fluorescence intensity of all 63 measured portions was determined,

and the fraction of all measurements with BRXL2 fluorescence intensity above half of this maximum was calculated as the BPI. BPIs of all the measured cells in the same conditions were then grouped, plotted, and compared with other conditions.

For the quantification of BRXL2 polarity persistence, the duration of BRXL2 polarity persistence was counted manually in different genotypes and treatments. For each cell, the beginning of post-ACD BRXL2 polarity was set as the image in which formation of the cell plate was first visible, while the end of post-ACD BRXL2 was set as the first time frame where BRXL2 polar crescent was no longer visually detectable. Post-ACD BRXL2 polarity dynamics from three individual cells of Col-0 and *ctr1*$^{G738R}$ (without added sugar) were measured with POME (*Gong et al., 2021*), and the dynamic of the BPI and the normalized crescent amplitude (peak amplitude divided by that at timepoint 0) were quantified and plotted in *Figure 6B* and *Figure 6—figure supplement 1A*.

For the quantification of SPCH reporter dynamics under mock, 10 µM ACC, or 2% glucose treatment, 10 random ACDs of each condition were chosen from respective time lapse of the genomic *pSPCH::gSPCH-YFP* reporter expressed in a *spch3* null mutant background (taken with the same basic imaging settings). Each ACD was then analyzed with the TrackMate FIJI plugin (*Tinevez et al., 2017*) for automated nucleus segmentation and tracking. Mean fluorescence intensity of the gSPCH-YFP reporter of each nucleus at each timepoint was then extracted and plotted in *Figure 6—figure supplement 1D*.

## Lineage tracing analysis

In each lineage tracing experiment, whole leaf images of the same leaves from different days were grouped together. Each individual cell from the 3 dpg (or any starting timepoint) was then annotated for divisions and lineage relationship each day after the starting point. BRXL2 reporter presence and polarity level in each cell and in different days was also recorded to help determine the division type as GMC divisions display depolarized BRXL2 while amplifying and spacing ACDs harbor a polarized BRXL2 crescent. After all the cells on the leaf epidermis were annotated, division behaviors of all cells are summarized in a spreadsheet, from which the total number and the fraction of each division type, fraction of each meristemoid behavior, total number of cells at the starting and ending timepoints, and any other developmental behaviors of interest were calculated.

## RNA extraction and qRT-PCR analysis

Whole seedlings of Col-0 and *hxk1-3* at 7 dpg were collected. Seven seedlings were grouped into a biological replicate, and two biological replicates per genotype were assayed. RNA of each biological replicate was then extracted with RNeasy plant mini kit (QIAGEN) with on-column DNAse digestion. Then, 1 µg of total RNA was used for cDNA synthesis using the iScript cDNA synthesis kit (Bio-Rad). The qPCR reactions were performed on a CFX96 Real-Time PCR detection system (Bio-Rad) with the Ssoadvanced Universal SYBR Green Supermix (Bio-Rad). Three technical replicates were performed per biological replicate. Expression level of *HXK1* was then calculated and normalized to the expression level of the reference gene *UBC21* using the ΔΔCT method.

## Statistical analysis

All statistical analyses in this paper were performed in RStudio. Unpaired Mann–Whitney tests and Student's *t*-test were conducted to compare two data samples with compare_means function from the ggpubr package (*Kassambara, 2020*). For all graphs, p-values from the unpaired Mann–Whitney tests or Student's *t*-test were directly labeled on these graphs except in *Figure 1E*, where values are provided in *Figure 1—figure supplement 3*.

## Primers

| Primer name | Purpose | Sequence |
| --- | --- | --- |
| CTR1_fw | Cloning | CACCATGGAAATGCCCGGTAGAAG |
| CTR1_rv | Cloning | CAAATCCGAGCGGTTGGGCGG |

*Continued on next page*

*Continued*

| Primer name | Purpose | Sequence |
| --- | --- | --- |
| I miR-s CTR1 | Cloning | GATATTTGATTTGACGCACGCAGTCTCTCTTTTGTATTCC |
| II miR-s CTR1 | Cloning | GACTGCGTGCGTCAAATCAAATATCAAAGAGAATCAATGA |
| III miR-s CTR1 | Cloning | GACTACGTGCGTCAATTCAAATTTCACAGGTCGTGATATG |
| IV miR-s CTR1 | Cloning | GAAATTTGAATTGACGCACGTAGTCTACATATATATTCCT |
| hxk1-3_LP | Genotyping | TTGATTATTTCTTCTTTCTGGCTTG |
| hxk1-3_RP | Genotyping | AGAACAGAAAACTGACATCTGAACC |
| *ein2-5*_fw | Genotyping | GCTCTTGTTCTTCTCTAGTC |
| *ein2-5*_rv | Genotyping | GAAGCATCATTGCCACCAAG |
| *ctr1*$^{G738R}$_fw | Genotyping/sequencing | CTAGGTCCTATTTCCAATGGAAG |
| *ctr1*$^{G738R}$_rv | Genotyping/sequencing | GGATTTAAGTTACCCCATGGTTG |
| UBC21_qPCR_fw | qRT-PCR | TCCTCTTAACTGCGACTCAGG |
| UBC21_qPCR_rw | qRT-PCR | GCGAGGCGTGTATACATTTG |
| HXK1_5′_qPCR_fw | qRT-PCR | CTGAATCCAGGCGAACAGA |
| HXK1_5′_qPCR_rw | qRT-PCR | TGTATCGCCAAAGAAAGCAG |
| HXK1_3′_qPCR_fw | qRT-PCR | AGCTACGTTGATAATCTTCCTTCC |
| HXK1_3′_qPCR_rw | qRT-PCR | TGTTTAACAACACGCTCTTGC |

## Acknowledgements

We thank Dr. Markus Grebe for providing the seeds of *ctr1-btk*, Dr. Wenrong He and Dr. Hongwei Guo for providing the seeds of *ein3eil1ebf1ebf2* and *EIN3ox*, Dr. Jiaying Zhu and Dr. Zhiyong Wong for providing the seeds of *gin2*, and Dr. Jose Alonso for providing the seeds of *ein2-5* and *ctr1-1*. We also thank Katelyn McKown, Gabriel Amador, and other members of the Bergmann lab for valuable feedback on the manuscript and Dr. José Dinneny for access to microscopy resources.

## Additional information

### Competing interests

Dominique C Bergmann: Reviewing editor, *eLife*. The other authors declare that no competing interests exist.

### Funding

| Funder | Grant reference number | Author |
| --- | --- | --- |
| Howard Hughes Medical Institute | | Yan Gong<br>Nidhi Sharma<br>Dominique C Bergmann |
| Swiss National Science Foundation | EPM-PBLAP3-142757 | Julien Alassimone |
| European Molecular Biology Organization | ALTF-878-2013 | Julien Alassimone |

The funders had no role in study design, data collection and interpretation, or the decision to submit the work for publication.

## Author contributions
Yan Gong, Conceptualization, Data curation, Formal analysis, Validation, Investigation, Visualization, Methodology, Writing - original draft, Writing - review and editing; Julien Alassimone, Conceptualization, Investigation; Rachel Varnau, Nidhi Sharma, Investigation; Lily S Cheung, Visualization, Methodology, Writing - review and editing; Dominique C Bergmann, Conceptualization, Data curation, Formal analysis, Supervision, Funding acquisition, Writing - original draft, Project administration, Writing - review and editing

## Author ORCIDs
Yan Gong https://orcid.org/0000-0003-1329-7096
Julien Alassimone https://orcid.org/0000-0002-8118-2605
Rachel Varnau https://orcid.org/0000-0002-3203-9597
Nidhi Sharma http://orcid.org/0000-0002-9725-5338
Lily S Cheung https://orcid.org/0000-0001-8089-7783
Dominique C Bergmann https://orcid.org/0000-0003-0873-3543

## Decision letter and Author response
Decision letter https://doi.org/10.7554/eLife.63335.sa1
Author response https://doi.org/10.7554/eLife.63335.sa2

## Additional files

### Supplementary files
• Transparent reporting form

### Data availability
All data generated on analyzed during this study are include in the manuscript and supporting files.

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
