## [Decision Letter]

**Acceptance summary:**

In this study, the authors identified a hitherto unknown connection between the polarity of BRXL2 in the stomatal lineage cells and ethylene signaling through a genetic screen. They quantified the polarity of individual cells during cell division across cell populations, and revealed that ethylene and glucose signaling, which are involved in environmental and nutritional pathways, respectively, regulate antagonistically the balance of asymmetric and symmetric cell divisions in the stomatal lineage. Furthermore, they discussed a potential link between BRXL2 stability in stomatal lineage ground cells and the behavior of their smaller meristematic sister cells (meristemoids). Overall, this study provides insights into understanding how environmental factors and nutritional status affect stem cell behavior in the stomatal lineage.

**Decision letter after peer review:**

Thank you for submitting your article "Tuning self-renewal in the *Arabidopsis* stomatal lineage by hormone and nutrient regulation of asymmetric cell division" for consideration by *eLife*. Your article has been reviewed by three peer reviewers, and the evaluation has been overseen by Hao Yu as the Reviewing Editor, and Christian Hardtke as the Senior Editor. The following individuals involved in review of your submission have agreed to reveal their identity: Olivier Hamant (Reviewer #1); Dolf Weijers (Reviewer #2).

The reviewers have discussed the reviews with one another and the Reviewing Editor has drafted this decision to help you prepare a revised submission.

Summary:

In this study, the authors identified a hitherto unknown connection between the polarity of BRXL2 in the stomatal lineage cells and ethylene signaling through a genetic screen. They quantified the polarity of individual cells during cell division across cell populations, and revealed that ethylene and glucose signaling, which are involved in environmental and nutritional pathways, respectively, regulate antagonistically the balance of asymmetric and symmetric cell divisions in the stomatal lineage.

Furthermore, they discussed a potential link between BRXL2 stability in stomatal lineage ground cells (SLGCs) and the behavior of their smaller meristematic sister cells (meristemoids). Overall, this study provides insights into understanding how environmental factors and nutritional status affect stem cell behavior in the stomatal lineage.

Essential revisions:

While there was a common interest about the topic and the relevant findings, some concerns on the underlying molecular mechanisms, quality of the experiments, and interpretation of results as listed below need to be addressed to support the conclusions.

1) Although there is evidence for the correlation between ethylene/glucose signalling and the degree of polarity of BRXL2 and stomatal production, the underlying mechanisms and the functional specificity of BRXL2 polarization/depolarization are unclear. We suggest the authors to examine other polarity proteins, such as BASL, in some key experiments, including analyzing their activities under ethylene/glucose influence and/or in the signaling mutants. In addition, investigation of the leaf response to ethylene, light, and sugar in various genetic backgrounds, such as *brxl2* and *basl* mutants or overexpression lines, will also help to clarify the role of BRXL2 and other polarity proteins in linking systemic signaling with stomatal stem-cell potential and leaf growth. These proposed experiments are necessary for evaluating the significance of the mechanisms underlying the correlation between systemic information (ethylene/glucose signalling) and stomatal lineage stem cell behavior.

2) Examination of glucose effect in Figure 5 could be improved by increasing sample size and/or optimizing the growth/genetic conditions under which the BPI baseline is higher.

3) Further analysis and clarification of the effects of ethylene/glucose treatment on SPCH protein level or its behavior are needed to support the current results in the manuscript, which seems different from the observations in a recent publication (Han et al., 2020). Similarly, the authors should explain the difference in the stomatal index of sugar-treated seedlings between this study and the work published by Han et al.

4) Please include a discussion on the potential mechanism(s) for controlling BRXL2 persistence at the molecular level.

Please also take into consideration the other specific comments from the reviewers below to revise the manuscript.

Reviewer #1:

The authors reveal a role of ethylene and glucose in determining BRXL2 polarity, with implication for the balance between symmetric and asymmetric cell division in the stomatal lineage, and thus organ size. Beyond the integration of the stomatal genetic pathway with biochemical cues (reflecting environmental conditions), they also find that BRXL2 polarity is a two-step process and that these above-mentioned cues rather play a role in the second step (polarity maintenance). The relation between glucose and ethylene is explored but remains open-ended.

This is a very interesting work, with a solid genetic backbone, and a key finding (link with biochemical cues and sequential polarity). Some loose ends need to be fixed before publication though. This may help clarify the main message too. In fact, the title, impact statement and some of the Abstract becomes clear only after one reads the paper. I hope the suggestions below will be helpful.

1) How specific is the BRXL2 depolarization ? Have the authors investigated other polarly localized proteins ? It would be important to have such a control, at least for the main/simpler tests (e.g. ACC and sugar treatments).

2) "To test if glucose regulates stomatal divisions through EIN3 inhibition, we compared BPIs in the ethylene insensitive mutants *ein2-5* and *ein3eil1ebf1ebf2* grown with and without glucose in the media. Both mutants had significantly reduced BPIs with 2% glucose at 4.5 dpg (Figure 5C and Figure 5—figure supplement 2A-F), suggesting that the influence of glucose in stomatal divisions is independent of the core components of ethylene signaling.” In fact the baseline in these mutants is already low without sugar, and the reduction after sugar treatment is not that significant (Figure 5C : p-value : 0.023 and 0.0095). So I would be more cautious here. Maybe one way to go around this is to increase sample size (i.e. >30 cells per genotype). Based on the results obtained on TOR and HXK1, I would not exclude a possible response saturation here, meaning that one would need to find growth/genetic conditions in which the baseline is higher, e.g. *ein2* in conditions X with a higher BPI, then treat with sugar – if the BPI remains high, sugar is acting via ethylene. Based on the authors' results, condition X could be low light actually. Of course it's too early to say, but if that works, it would also help showcase the result in Figure 5F which is a bit buried and would deserve more light.

3) The conclusion about a possible overarching signal (biochemical or mechanical) is consistent with the idea that ethylene and glucose may affect the whole cell physiology, that the maintenance of BRXL2 polarity is affected (not the initial trigger) and possibly that BRXL2 might only be one of the many targets for which an impact on polarity is found. This relates to comment 1 on the need to test at least one or two other polar proteins to check for specificity. This would not decrease the impact of the article if all proteins respond, it would simply be consistent with a more generic response (which is equally interesting). A possible "generic" scenario is that ethylene might promote wall differentiation (hence reduced cell size), thus reduce wall tension, thus hindering polarity. Glucose (maybe via its antagonistic role on ethylene, see comment 2) would have a counteracting effect. This actually would be very much in line with Brinkmann and Bergmann, 2017, from the same group. This could also be discussed in relation to turgor pressure (see Long 2020 Curr Biol) and tested (e.g. cellulose or isoxaben treatment would weaken the wall and might restore BRXL2 polarity in *ctr1*)

Reviewer #2:

Gong et al. describe a very careful analysis of BRXL2 polarization in wild-type and ctr mutant stomatal lineage cells. In doing so, they find a clear correlation between persistence of the polar BRXL2 crescent after asymmetric division and the fate of the daughter cell. Longer persistence of the crescent correlates with increased stemness. Through analyzing the ctr mutant, ethylene is identified as a trigger that influences stomatal index (and leaf size) by influencing post ACD stemness. Conversely, glucose signaling (through yet unidentified components) also signals to extend stemness by promoting persistence of BRXL2.

This work presents a very interesting paradigm of control of organ growth and cellular composition by (environmental) signals through tuning stemness. It is a bit disappointing however, that much is based on correlations, and no clear molecular mechanism is provided. Is tuning BRXL2 levels sufficient for this response? Is only BRXL2 controlled, or would BASL likely be a target of this regulation?

There are a few points that could help in addressing these questions:

1) Is the leaf growth response to ethylene, light, sugars altered in the various *brxl2* and *basl* mutants or overexpression lines?

2) Given that BRXL2 and BASL are interdependent in their function and localization, it would be interesting to determine the behavior of BASL in a similar set of assays.

3) It is not clear to me how the authors envisage that BRXL2 persistence is controlled at the molecular level. Does this involve active stabilization (or active degradation in the case where BXRL2 does not persist)?

Reviewer #3:

In this manuscript from the Bergmann lab, the authors report the regulation of a polarity factor during stomatal asymmetric cell divisions (ACDs) and the involvement of ethylene and glucose signaling in the process. Through a genetic screen, they identified a surprising connection between the polarity of BRXL2 in the stomatal lineage cells and ethylene signaling. Lineage tracing of WT and the *ctr1* mutant suggested that the amplifying ACDs are suppressed in the ethylene signaling mutant. They further showed that ethylene and glucose signalling can oppositely influence the degree of polarity of BRXL2 and stomatal production. Finally, they described a potential connection between BRXL2 stability in the stomatal lineage ground cells (SLGCs) and the behaviour of their smaller meristematic sister cells (meristemoids).

The correlation between the degree of polarity of a polarized factor and the stomatal stem cell activity, is novel, and could represent a broader phenomenon that was previously overlooked. The quantitative and lineage tracing approaches also represent powerful methods to describe polarity and dissect the activity of the stomatal lineage. However, there are some additional experiments that would help strengthen the authors' claims.

1) The authors exclusively used BRXL2 as the polarity protein in their analyses. What about the behaviour of BASL, the other critical component of the polarity complex? Analyzing the activity of BASL under ethylene/glucose influence or in the signaling mutants would provide a fuller picture of the phenomenon and may help dissect the underlying mechanism.

2) It is somewhat surprising that SPCH protein level or its behaviour does not seem to be influenced by ethylene/glucose treatment, especially since a recent study proposes that SPCH protein can be stabilized by sugar (Han et al., 2020). How many times were the time-lapse analyses of SPCH in Figure 6—figure supplement 1 conducted? Additional quantitative data on the persistence/peak levels of SPCH during ACD, similar to how BRXL2 was characterized, would provide better support of the authors' proposal. Further, during ACD, how is the protein level of BRXL2 in WT compared to *ctr1*?

3) Stomatal index is a useful but relatively indirect measure of the activity of the stomatal lineage, and thus, the lineage tracing approach that the authors employed is superior in this context. However, since SI still represents developmental outcome, it is again surprising to see that the sugar-treated seedlings here had a significantly lower SI compared to WT, while they were higher or indistinguishable in the previous study (Han et al., 2020). How can this be explained?

4) The authors may want to cite Serna and Fenoll, 1996, as an earlier study on the effect of ethylene on stomatal development.

5) For Figure 3G, I found it confusing to include GMC SCD here, as opposed to having only the two types of ACDs, which would represent the behavior of meristemoids. Also, the two types of ACDs can also be shown in Figure 3F to make it more useful.

6) What is the sample size for Figure 3I?

7) For Figure 7B, it would be more appropriate to include a "question mark" on arrows/text that do not have experimental support.

[Editors' note: further revisions were suggested prior to acceptance, as described below.]

Thank you for submitting your revised article "Tuning self-renewal in the *Arabidopsis* stomatal lineage by hormone and nutrient regulation of asymmetric cell division" for consideration by *eLife*. Your article has been reviewed by three peer reviewers, and the evaluation has been overseen by Hao Yu as the Reviewing Editor, and Jürgen Kleine-Vehn as the Senior Editor. The following individuals involved in review of your submission have agreed to reveal their identity: Olivier Hamant (Reviewer #2); Dolf Weijers (Reviewer #3).

The reviewers were very enthusiastic about your work and the Reviewing Editor has drafted this to help you prepare a revised submission.

Essential Revisions:

1) For Figure 1—figure supplement 2B, the second and third BRXL2-YFP images of Col-0 seems to be mixed up. The authors should check these images and correct them if necessary.

---

## [Author Response]

Essential revisions:While there was a common interest about the topic and the relevant findings, some concerns on the underlying molecular mechanisms, quality of the experiments, and interpretation of results as listed below need to be addressed to support the conclusions.1) Although there is evidence for the correlation between ethylene/glucose signalling and the degree of polarity of BRXL2 and stomatal production, the underlying mechanisms and the functional specificity of BRXL2 polarization/depolarization are unclear. We suggest the authors to examine other polarity proteins, such as BASL, in some key experiments, including analyzing their activities under ethylene/glucose influence and/or in the signaling mutants. In addition, investigation of the leaf response to ethylene, light, and sugar in various genetic backgrounds, such as brxl2 and basl mutants or overexpression lines, will also help to clarify the role of BRXL2 and other polarity proteins in linking systemic signaling with stomatal stem-cell potential and leaf growth. These proposed experiments are necessary for evaluating the significance of the mechanisms underlying the correlation between systemic information (ethylene/glucose signalling) and stomatal lineage stem cell behavior.

We thank the reviewers for the suggestion of expanding the study to ask not just whether BRXL2 polarity reported a response to external conditions, but whether it (and other polarity proteins) were part of the mechanism mediating that response. We have now added data on BASL polarity in stomatal lineage cells. These data are from seedlings harboring a BASLp::YFP-BASL reporter grown under two treatments (+ control): germination on ½ MS plate with and without 2% glucose or 10µM ACC. We found that ACC treatment significantly reduced tissue-level BASL polarity, while glucose treatment significantly enhanced it (Figure 4—figure supplement 2B-E). This was similar to the result with BRXL2 polarity. These results suggest that glucose and ethylene’s effect is not just on BRXL2, but stomatal cell polarity in general.

We then tested whether stomatal lineage development can be modulated by ethylene signaling pathway in the absence of the polarity module. BRXL2 is redundant with other BRX family members and our genetic data in Rowe et al., 2019 indicate that BRX-family and BASL work together, so to get the clearest readout of the effect of loss of polarity on the response to ACC, we looked at the *basl* null mutant (*basl-2*).

In this cell polarity mutant, we found that 10µM ACC treatment can still increase the stomatal index (SI) (Figure 4—figure supplement 2F). The degree of increase of SI in *basl-2* upon ACC treatment is less than that in wild-type Col-0, but the starting SI is also higher. Given this result, it appears that part, but not all, of ethylene’s regulation on stomatal lineage development might be through the polarity protein complex. Our models at the end discuss how polarity proteins could be a driver or a reporter of cell fate decisions.

It was not clear to us whether the suggestion to look at overexpression of polarity factors was to test whether polarization in other cells could be modulated, or whether the phenotype of the OE lines with respect to stomatal production was to be measured. Because the temporal persistence of BRXL2 polarity in stomatal lineage cells appears to be the relevant behavior for division response, we felt that examining polarity in ectopic positions would not be biologically meaningful. OE (or expression of hyperactive) polarity proteins does not lead to noticeable phenotypes and/or we haven’t been able to observe the reported phenotypes in our hands. .

2) Examination of glucose effect in Figure 5 could be improved by increasing sample size and/or optimizing the growth/genetic conditions under which the BPI baseline is higher.

As noted by the reviewers, the BPI “baseline” in cotyledons from ethylene insensitive mutants *ein2-5* and *ein3eil1ebf1ebf2* is lower than for Col-0, and therefore it is possible that we would not be able to see a mild effect of glucose treatment.

We had this concern as well, and so we measured the ability of glucose to promote BRXL2 polarity in *ein2-5, ein3eil1ebf1ebf2*, and Col-0 cotyledons at a later developmental stage (Figure 5C, 4.5 dpg), than we did for most other measurements of BPI (4dpg, compare Col-0 BPI in Figure 1E or 2J to that in 5C). At this slightly later timepoint, BPI are no longer saturated (100% less than 0.3) in *ein2-5* or *ein3eil1ebf1ebf2* and we observed that glucose significantly promoted tissue level BRXL2 polarity (Figure 5—figure supplement 2A-F) and reduced BPI in both *ein2-5* and *ein3eil1ebf1ebf2* (Figure 5C).

To push the BPI baseline even higher, we imaged the BRXL2 reporter at 9 dpg in 11 uE low light condition (same experimental procedure as the experiment of Figure 5F) and found that the addition of 2% glucose significantly promoted tissue level BRXL2 polarity in both *ein3eil1ebf1ebf2* and Col-0 true leaves (Figure 5—figure supplement 5). These new results reinforce the idea that the regulation of BRXL2 polarity and stomatal lineage development by glucose is unlikely through the cross talk between glucose signaling and ethylene signaling.

3) Further analysis and clarification of the effects of ethylene/glucose treatment on SPCH protein level or its behavior are needed to support the current results in the manuscript, which seems different from the observations in a recent publication (Han et al., 2020). Similarly, the authors should explain the difference in the stomatal index of sugar-treated seedlings between this study and the work published by Han et al.

The reviewers are correct that it appears that our findings on SPCH differ from those reported in Han et al., 2020. We have now stated explicitly that our experimental set-up and analysis differs significantly from Han et al., 2020. But, we also note that in the few places where we do similar experiments, our results trend in the same direction. The main differences are:

1) Different growth conditions

The majority of experiments conducted by Han et al., 2020, were from seedlings grown in MS liquid culture while our experiments were from seedlings grown on solidified agar-MS plates. Epidermal development is severely impacted when Arabidopsis seedlings are grown in the liquid culture (Author response image 1), likely due to hypoxia and stress. To illustrate the difference between plants grown in liquid culture and on plates, we imaged the whole cotyledons from these two conditions, quantified the surface area of these cotyledons, and counted the number of cells on the abaxial epidermis from each cotyledon (Author response image 1). We found that at the same developmental stages, cotyledons grown on plates are about three times bigger in leaf size and have about two times more cells on the abaxial epidermis, suggesting that both cell division and differentiation of stomatal lineage are strongly suppressed in seedlings grown in liquid culture.

**Author response image 1. respfig1:** Liquid culture significantly alters epidermal development. (**A**) Confocal images of 4 dpg Col-0 cotyledons grown on regular ½ MS plates (top) and in ½ MS liquid culture (bottom). Cell outline is visualized with *pATML1::RCI2A-mCherry* (gray). (**B-C**) Quantification of cotyledon surface area (**B**) and total cell numbers (**C**) from cotyledons shown in A. All p-values are calculated by Student’s *t*- test. Scale bars in A, 50 μm.

2) Different SPCH reporter

We monitored a SPCH reporter that contains the full genomic regions (gSPCH, Lopez-Anido et al., 2020), that shows a greater dynamic range than the older SPCH CDS reporter (cSPCH) used in the Han paper. As a result, we believe our experiments are more sensitive in detecting the full picture of SPCH expression profile in the stomatal lineage.

3) Different measurements of SPCH, and different sugar treatments

It is also important to note that changes in SPCH levels could be measured as an increase in the number of cells expressing SPCH or in the intensity of SPCH in those cells that express it. Figure 1 from Han et al., 2020, showed a higher percentage of cells are expressing SPCH, not that SPCH levels are higher per cell, when 1% sucrose was added to the ½ MS liquid culture. Judging from the Figure 1 from Han et al., cSPCH reporter fluorescence intensity under mock and 1% sucrose treatment, they reach a similar conclusion as our findings from Figure 6 F-I.

To test whether sucrose and glucose have different effects on SPCH expression, we imaged and quantified the gSPCH reporter from *Arabidopsis* seedlings grown on ½ MS plate with no sugar, 2% glucose, 2% sucrose, and 10uM ACC (Author response image 2). Similar to glucose, sucrose didn’t increase the fluorescence intensity of gSPCH reporter in SPCH expressing cells.

**Author response image 2. respfig2:** SPCH reporter fluorescence intensity under different sugar and ethylene treatments. Quantification of gSPCH reporter fluorescence intensity at 4 dpg in mock, 10 μM ACC, 2% glucose, and 2% sucrose treated Col-0 cotyledons (n = 8 cotyledons/treatment; n>200 cells/treatment).

To characterize SPCH dynamics during ACDs under different ethylene and sugar treatments, we performed time-lapse imaging of the gSPCH reporter in each of these conditions and quantified the changes of SPCH reporter intensity in individual ACDs (10 ACDs per condition). There is variation in SPCH expression in each measured cell, but no clear variation that correlates with ethylene or sugar treatment (Figure 6—figure supplement 1C-D).

Together, these results suggest SPCH itself is likely not the direct target of glucose or ethylene signaling during relatively unstressed conditions. Rather, we contend that SPCH provides a downstream readout of changes in meristemoid division behaviors, since SPCH is required for ACDs. With more meristemoids undergoing ACDs in glucose treated cotyledons and fewer meristemoids undergoing ACDs in ACC treated cotyledons, the number of SPCH expressing cells are expected to be different.

Stomatal index is the readout of the combination of stomatal lineage entry divisions in meristemoid mother cells, amplifying division in meristemoids, and spacing division in SLGCs, and it has been reported to be sensitive to environmental changes. Due to the strong inhibition of stomatal lineage development in liquid culture (Author response image 1), it is reasonable to see stomatal index to respond differently to sugar treatments.

4) Please include a discussion on the potential mechanism(s) for controlling BRXL2 persistence at the molecular level.Please also take into consideration the other specific comments from the reviewers below to revise the manuscript.

We think BRXL2 persistence in SLGC could either be 1) a readout of meristemoids self-renewing capacity though the EPF2-MAPK-BASL polarity pathway or 2) an active source of intercellular signaling communication or mechanical force originated in SLGC that regulates meristemoids ability to undergo amplifying divisions. We added discussion of these points in the Discussion section of the manuscript and Figure 7 and briefly describe the models below.

In the first case, EPF2 secreted by the actively self-renewing meristemoid is perceived by the neighboring SLGC, elevating SLGC’s EPF2-MAPK signaling (Lee et al., 2015). This elevated MAPK activity then promotes export of nuclear BASL to the polar crescent, sustaining BASL/BRXL2 polarity (Zhang et al., 2016; Zhang et al., 2015). When the meristemoid transitions to a GMC, EPF2 is no longer produced, resulting in lower MAPK activity in the neighboring SLGC. Replenishment of cortical BASL from the nuclear pool is then lost, leading BASL/BRXL2 polarity to diminish.

In the second case, because of the intense cross-regulation between BASL/BRXL2 polar crescent and the kinases (MAPKs, PAX, BIN2, other AGC kinases etc.) they scaffold (Houbaert et al., 2018; Marhava et al., 2020; Zhang et al., 2015), BASL/BRXL2 polar crescent could act through these kinase pathways to regulate the transport and local distribution of auxin or brassinosteroids, Both auxin or brassinosteroid have been shown to regulate stem-cell division and lineage progression in the stomatal lineage (Houbaert et al., 2018; Kim et al., 2012; Le et al., 2014) and change in their local concentration is likely to alter the development of the surrounding cells.

Alternatively, the orientation of BASL/BRXL2 polar crescent has been linked to tissue level polarity and proposed to regulate local cell expansion (Bringmann and Bergmann, 2017; Mansfield et al., 2018). Prolonged BASL/BRXL2 could potentially trigger redistribution of regional mechanical force and affect the stem-cell divisions in the surrounding cells.

Reviewer #1:The authors reveal a role of ethylene and glucose in determining BRXL2 polarity, with implication for the balance between symmetric and asymmetric cell division in the stomatal lineage, and thus organ size. Beyond the integration of the stomatal genetic pathway with biochemical cues (reflecting environmental conditions), they also find that BRXL2 polarity is a two-step process and that these above-mentioned cues rather play a role in the second step (polarity maintenance). The relation between glucose and ethylene is explored but remains open-ended.This is a very interesting work, with a solid genetic backbone, and a key finding (link with biochemical cues and sequential polarity). Some loose ends need to be fixed before publication though. This may help clarify the main message too. In fact, the title, impact statement and some of the Abstract becomes clear only after one reads the paper. I hope the suggestions below will be helpful.1) How specific is the BXL2 depolarization ? Have the authors investigated other polarly localized proteins ? It would be important to have such a control, at least for the main/simpler tests (e.g. ACC and sugar treatments).

Please see response to essential revision 1.

2) "To test if glucose regulates stomatal divisions through EIN3 inhibition, we compared BPIs in the ethylene insensitive mutants ein2-5 and ein3eil1ebf1ebf2 grown with and without glucose in the media. Both mutants had significantly reduced BPIs with 2% glucose at 4.5 dpg (Figure 5C and Figure 5—figure supplement 2A-F), suggesting that the influence of glucose in stomatal divisions is independent of the core components of ethylene signaling.” In fact the baseline in these mutants is already low without sugar, and the reduction after sugar treatment is not that significant (Figure 5C : p-value : 0.023 and 0.0095). So I would be more cautious here. Maybe one way to go around this is to increase sample size (i.e. >30 cells per genotype). Based on the results obtained on TOR and HXK1, I would not exclude a possible response saturation here, meaning that one would need to find growth/genetic conditions in which the baseline is higher, e.g. ein2 in conditions X with a higher BPI, then treat with sugar – if the BPI remains high, sugar is acting via ethylene. Based on the authors' results, condition X could be low light actually. Of course it's too early to say, but if that works, it would also help showcase the result in Figure 5F which is a bit buried and would deserve more light.

Please see response to essential revision 2.

3) The conclusion about a possible overarching signal (biochemical or mechanical) is consistent with the idea that ethylene and glucose may affect the whole cell physiology, that the maintenance of BRXL2 polarity is affected (not the initial trigger) and possibly that BRXL2 might only be one of the many targets for which an impact on polarity is found. This relates to comment 1 on the need to test at least one or two other polar proteins to check for specificity. This would not decrease the impact of the article if all proteins respond, it would simply be consistent with a more generic response (which is equally interesting). A possible "generic" scenario is that ethylene might promote wall differentiation (hence reduced cell size), thus reduce wall tension, thus hindering polarity. Glucose (maybe via its antagonistic role on ethylene, see comment 2) would have a counteracting effect. This actually would be very much in line with Bringmann and Bergmann, 2017, from the same group. This could also be discussed in relation to turgor pressure (see Long 2020 Curr Biol) and tested (e.g. cellulose or isoxaben treatment would weaken the wall and might restore BRXL2 polarity in ctr1).

Please see response to essential revision 4.

Reviewer #2:Gong et al. describe a very careful analysis of BBRXL2 polarization in wild-type and ctr mutant stomatal lineage cells. In doing so, they find a clear correlation between persistence of the polar BRXL2 crescent after asymmetric division and the fate of the daughter cell. Longer persistence of the crescent correlates with increased stemness. Through analyzing the ctr mutant, ethylene is identified as a trigger that influences stomatal index (and leaf size) by influencing post ACD stemness. Conversely, glucose signaling (through yet unidentified components) also signals to extend stemness by promoting persistence of BRXL2.This work presents a very interesting paradigm of control of organ growth and cellular composition by (environmental) signals through tuning stemness. It is a bit disappointing however, that much is based on correlations, and no clear molecular mechanism is provided. Is tuning BRXL2 levels sufficient for this response? Is only BRXL2 controlled, or would BASL likely be a target of this regulation?There are a few points that could help in addressing these questions:1) Is the leaf growth response to ethylene, light, sugars altered in the various brxl2 and basl mutants or overexpression lines?

Please see response to essential revision 1.

2) Given that BRXL2 and BASL are interdependent in their function and localization, it would be interesting to determine the behavior of BASL in a similar set of assays.

Please see response to essential revision 1.

3) It is not clear to me how the authors envisage that BRXL2 persistence is controlled at the molecular level. Does this involve active stabilization (or active degradation in the case where BXRL2 does not persist)?

Please see response to essential revision 4.

Reviewer #3:In this manuscript from the Bergmann lab, the authors report the regulation of a polarity factor during stomatal asymmetric cell divisions (ACDs) and the involvement of ethylene and glucose signaling in the process. Through a genetic screen, they identified a surprising connection between the polarity of BRXL2 in the stomatal lineage cells and ethylene signaling. Lineage tracing of WT and the ctr1 mutant suggested that the amplifying ACDs are suppressed in the ethylene signaling mutant. They further showed that ethylene and glucose signalling can oppositely influence the degree of polarity of BRXL2 and stomatal production. Finally, they described a potential connection between BRXL2 stability in the stomatal lineage ground cells (SLGCs) and the behaviour of their smaller meristematic sister cells (meristemoids).The correlation between the degree of polarity of a polarized factor and the stomatal stem cell activity, is novel, and could represent a broader phenomenon that was previously overlooked. The quantitative and lineage tracing approaches also represent powerful methods to describe polarity and dissect the activity of the stomatal lineage. However, there are some additional experiments that would help strengthen the authors' claims.1) The authors exclusively used BRXL2 as the polarity protein in their analyses. What about the behaviour of BASL, the other critical component of the polarity complex? Analyzing the activity of BASL under ethylene/glucose influence or in the signaling mutants would provide a fuller picture of the phenomenon and may help dissect the underlying mechanism.

Please see response to essential revision 1.

2) It is somewhat surprising that SPCH protein level or its behaviour does not seem to be influenced by ethylene/glucose treatment, especially since a recent study proposes that SPCH protein can be stabilized by sugar (Han et al., 2020). How many times were the time-lapse analyses of SPCH in Figure 6—figure supplement 1 conducted? Additional quantitative data on the persistence/peak levels of SPCH during ACD, similar to how BRXL2 was characterized, would provide better support of the authors' proposal. Further, during ACD, how is the protein level of BRXL2 in WT compared to ctr1?

Please see response to essential revision 3.

3) Stomatal index is a useful but relatively indirect measure of the activity of the stomatal lineage, and thus, the lineage tracing approach that the authors employed is superior in this context. However, since SI still represents developmental outcome, it is again surprising to see that the sugar-treated seedlings here had a significantly lower SI compared to WT, while they were higher or indistinguishable in the previous study (Han et al., 2020). How can this be explained?

Please see response to essential revision 3.

4) The authors may want to cite Serna and Fenoll, 1996, as an earlier study on the effect of ethylene on stomatal development.

We added the citation in the “Ethylene signaling regulates polarity protein complex and stomatal lineage development” section.

5) For Figure 3G, I found it confusing to include GMC SCD here, as opposed to having only the two types of ACDs, which would represent the behavior of meristemoids. Also, the two types of ACDs can also be shown in Figure 3F to make it more useful.

The SCD/ACD ratio concept that we introduce early on and use throughout the text groups all observed ACDs together. Thus we think it’s important to give the reader a graphical representation of this information (Figure 3F). Figure 3G is designed to report the shift in types of divisions in the *ctr1* mutant. We think it is useful to demonstrate that even with an overall reduction in the number of cell divisions (Figure 3F) not all division types are affected equally. At the stages we are able to follow, most entry divisions would have already occurred and thus they would represent a very small # of total divisions.

6) What is the sample size for Figure 3I?

N=3. We added this info to the figure legend of Figure 3I.

7) For Figure 7B, it would be more appropriate to include a "question mark" on arrows/text that do not have experimental support.

We added the question mark as suggested.

[Editors' note: further revisions were suggested prior to acceptance, as described below.]

Essential Revisions:1) For Figure 1—figure supplement 2B, the second and third BRXL2-YFP images of Col-0 seems to be mixed up. The authors should check these images and correct them if necessary.

Thank you for catching this. The images were indeed shifted. We have fixed this and made sure this (and other cells) are now correctly aligned.

Reference:

Bringmann, M., & Bergmann, D. C. (2017). Tissue-wide Mechanical Forces Influence the Polarity of Stomatal Stem Cells in Arabidopsis. *Curr Biol*,*27*(6), 877-883. https://doi.org/10.1016/j.cub.2017.01.059

Dow, G. J., Berry, J. A., & Bergmann, D. C. (2017). Disruption of stomatal lineage signaling or transcriptional regulators has differential effects on mesophyll development, but maintains coordination of gas exchange. *New Phytol*, *216*(1), 69-75. https://doi.org/10.1111/nph.14746

Han, C., Liu, Y., Shi, W., Qiao, Y., Wang, L., Tian, Y., Fan, M., Deng, Z., Lau, O. S., De Jaeger, G., & Bai, M. Y. (2020). KIN10 promotes stomatal development through stabilization of the SPEECHLESS transcription factor. *Nat Commun*, *11*(1), 4214. https://doi.org/10.1038/s41467-020-18048-w

Houbaert, A., Zhang, C., Tiwari, M., Wang, K., de Marcos Serrano, A., Savatin, D. V., Urs, M. J., Zhiponova, M. K., Gudesblat, G. E., Vanhoutte, I., Eeckhout, D., Boeren, S., Karimi, M., Betti, C., Jacobs, T., Fenoll, C., Mena, M., de Vries, S., De Jaeger, G., & Russinova, E. (2018). POLAR-guided signalling complex assembly and localization drive asymmetric cell division. *Nature*, *563*(7732), 574-578. https://doi.org/10.1038/s41586-018-0714-x

Kim, T. W., Michniewicz, M., Bergmann, D. C., & Wang, Z. Y. (2012). Brassinosteroid regulates stomatal development by GSK3-mediated inhibition of a MAPK pathway. *Nature*, *482*(7385), 419-422. https://doi.org/10.1038/nature10794

Le, J., Liu, X. G., Yang, K. Z., Chen, X. L., Zou, J. J., Wang, H. Z., Wang, M., Vanneste, S., Morita, M., Tasaka, M., Ding, Z. J., Friml, J., Beeckman, T., & Sack, F. (2014). Auxin transport and activity regulate stomatal patterning and development. *Nat Commun*, *5*, 3090. https://doi.org/10.1038/ncomms4090

Lee, J. S., Hnilova, M., Maes, M., Lin, Y. C., Putarjunan, A., Han, S. K., Avila, J., & Torii, K. U. (2015). Competitive binding of antagonistic peptides fine-tunes stomatal patterning. *Nature*, *522*(7557), 439-443. https://doi.org/10.1038/nature14561

Lopez-Anido, C. B., Vatén, A., Smoot, N. K., Sharma, N., Guo, V., Gong, Y., Anleu Gil, M. X., Weimer, A. K., & Bergmann, D. C. (2020). Single-Cell Resolution of Lineage Trajectories in the Arabidopsis Stomatal Lineage and Developing Leaf. *bioRxiv*, 2020.2009.2008.288498. https://doi.org/10.1101/2020.09.08.288498

Mansfield, C., Newman, J. L., Olsson, T. S. G., Hartley, M., Chan, J., & Coen, E. (2018). Ectopic BASL Reveals Tissue Cell Polarity throughout Leaf Development in Arabidopsis thaliana. *Curr Biol*, *28*(16), 2638-2646 e2634. https://doi.org/10.1016/j.cub.2018.06.019

Marhava, P., Aliaga Fandino, A. C., Koh, S. W. H., Jelinkova, A., Kolb, M., Janacek, D. P., Breda, A. S., Cattaneo, P., Hammes, U. Z., Petrasek, J., & Hardtke, C. S. (2020). Plasma Membrane Domain Patterning and Self-Reinforcing Polarity in Arabidopsis. *Dev Cell*, *52*(2), 223-235 e225. https://doi.org/10.1016/j.devcel.2019.11.015

Rowe, M. H., Dong, J., Weimer, A. K., & Bergmann, D. C. (2019). A Plant-Specific Polarity Module Establishes Cell Fate Asymmetry in the Arabidopsis Stomatal Lineage. *bioRxiv*, 614636. https://doi.org/10.1101/614636

Zhang, Y., Guo, X., & Dong, J. (2016). Phosphorylation of the Polarity Protein BASL Differentiates Asymmetric Cell Fate through MAPKs and SPCH. *Curr Biol*, *26*(21), 2957-2965. https://doi.org/10.1016/j.cub.2016.08.066

Zhang, Y., Wang, P., Shao, W., Zhu, J. K., & Dong, J. (2015). The BASL polarity protein controls a MAPK signaling feedback loop in asymmetric cell division. *Dev Cell*, *33*(2), 136-149. https://doi.org/10.1016/j.devcel.2015.02.022